

# Value of seasonal flow forecasts for enhancing reservoir operation and drought management in South Korea

Yongshin Lee[1], Andres Peñuela[2], Francesca Pianosi[1], Miguel Angel Rico-Ramirez[1]

[1] School of Civil, Aerospace and Design Engineering, University of Bristol, Bristol, BS8 1TR, UK

[2] Department of Agronomy, Unidad de Excelencia María de Maeztu, University of Cordoba, Cordoba, 14071, Spain

**Correspondence:** Yongshin Lee (yongshin.lee@bristol.ac.uk)

**Abstract.**

*Drought poses significant challenges across various sectors such as agriculture, water resources, environment, and energy. In the past few decades, numerous devastating droughts have been reported worldwide including in South Korea. A recent drought in South Korea, which lasted from 2013 to 2016, led to significant consequences including water restrictions and nationwide crop failures. Historically, reservoirs have played a crucial role in mitigating hydrological droughts by ensuring water supply stability. With exacerbating intensity and frequency of droughts attributed to climate change, enhancing the operational efficiency of existing reservoirs for drought management becomes increasingly important. This study examines the value of Seasonal Flow Forecasts (SFFs) in informing reservoir operations, focusing on two critical reservoir systems in South Korea. We assess and compare the value derived from using two deterministic scenarios (worst and 20-year return period drought) and two ensemble forecasts products (SFFs and Ensemble Streamflow Prediction, ESP). Our study proposes an innovative method for assessing forecast value, providing a more intuitive and practical understanding by directly relating it to the likelihood of achieving better operational outcomes compared to historical operation. Furthermore, we analyse the sensitivity of forecast value to key choices in the set-up of the simulation experiments. Our findings indicate that while deterministic scenarios show higher accuracy, forecast-informed operations with ensemble forecasts tend to yield greater value. This highlights the importance of considering the uncertainty of flow forecasts in operating reservoirs. Although SFFs generally show higher accuracy than ESP, the difference in value between these two ensemble forecasts is found to be negligible. Last, the sensitivity analysis shows that the method used to select a compromise release schedule between competing operational objectives is a key determinant of forecast value, implying that the benefits of using seasonal forecasts may vary widely depending on how priorities between objectives are established.*

**Keywords:** drought mitigation, reservoir operations, seasonal weather forecasts, seasonal flow forecasts, ensemble streamflow prediction, multi-objective optimisation, multi-criteria decision-making

## 1. Introduction

Drought stands as one of the major natural disasters with devastating impacts for various sectors including agriculture, water resources, environment, and energy (Mishra and Singh, 2010; Schwalm et al., 2017; Zhang et al., 2022). The severity of droughts is anticipated to escalate in the future under a warmer climate, but there is plenty of evidence to suggest that this increase may already be underway (Sheffield et al., 2012). In South Korea, a severe drought event, prolonged from 2013 to 2016 (Nam et al., 2015; Kwon et al., 2016), caused substantial consequences, such as water restrictions in certain regions and nationwide crop failures (K-water, 2018). Reservoirs have played a crucial role in mitigating drought impacts by stabilizing water supply and compensating for hydrological variability (Goldsmith and Hildyard, 1984). However, the increasing frequency and intensity of extreme droughts are posing greater challenges for reservoir operators (Sheffield et al., 2012; Schwalm et al., 2017). On the other hand, the construction of new reservoirs has become increasingly controversial in many countries, including South Korea, mainly due to concerns about the socio-economic costs and undesirable environmental impacts of reservoir development (Ehsani et al., 2017). This highlights the growing significance of enhancing the operation of existing reservoirs to mitigate drought damages. A key contribution to this end may come by improving flow forecasting systems and their use in support of decision-making under extreme weather conditions (Turner et al., 2017).

Advancements in numerical weather prediction systems over the past decade have significantly improved forecasting performance at longer time scales (Bauer et al., 2015; Alley et al., 2019). Seasonal weather forecasts, which provide predictions of weather variables (e.g., precipitation, temperature) for the next several months, have gained interest among researchers for their potential in enhancing water resources management. Accordingly, numerous studies have been conducted to transform seasonal weather forecasts into more relevant Seasonal Flow Forecasts (SFFs) across various regions of the world (e.g., Prudhomme et al., 2017; Arnal et al., 2018; Greuell et



al., 2018; Lucatero et al., 2018; Hurkmans et al., 2023). In many countries, however, practical reservoir operations still make limited use of SFFs. Even when water resource modelling is used to inform operational decisions, reservoir operators tend to run these models against deterministic scenarios such as the worst-case scenario (Yoe, 2019) or against Ensemble Streamflow Prediction (ESP) (Day, 1985). The worst-case scenario mimics the most extreme historical event to hedge risks associated with uncertainties in water management, whereas ESP generates an ensemble of flow forecasts by forcing a hydrological model with historical meteorological observations (Baker et al., 2021). Previous studies have identified as primary obstacles to the use of SFFs by water managers their tendency to adopt a risk-adverse approach (Block, 2011), the lack of experience in handling SFFs products, and the perceived low reliability of SFFs (Millner and Washington, 2011; Soares and Dessai, 2016; Jackson-Blake et al., 2022). Indeed, previous studies have shown that SFFs provide more accurate forecast than ESP only for the first or second months ahead (Yossef et al., 2013; Crochemore et al., 2016; Lucatero et al., 2018), and their performance decreases with increasing lead time (Greuell et al., 2018; Pechlivanidis et al., 2020).

In perspective of reservoir operations, however, more than the forecast accuracy, i.e. how well hydrological forecasts replicate observations, the attention should be directed to the forecast value, i.e., the benefits of using forecasts to inform operational decisions (Turner et al., 2017; Peñuela et al., 2020). Assessing forecast value may reveal situations where using forecasts enhances water management even if the accuracy is relatively low (Rougé et al., 2023). With this idea in mind, several studies have utilised model simulations to assess how using SFFs could have improved reservoir operations during past events. To achieve this, these studies have generated SFFs and fed them into a reservoir operation optimisation model to reconstruct and find the "optimal" release schedule for the given event, then assessed the effects by simulating against actual observed flows. The performances of such forecast-informed operations are then summarised through a set of performance indicators and compared to the performances obtained with a benchmark approach, such as optimising against a deterministic scenario or ESP, or using reservoir operation rules. The increase in performance with respect to the benchmark is regarded as the value of the SFFs. For example, Chiew et al. (2003) investigated the value of SFFs for agricultural supply from a reservoir in Australia, a region affected by El Nino/Southern Oscillation (ENSO) teleconnections. Their findings indicated that release scheduling informed by SFFs can yield modest benefits compared to using a predefined reservoir operation rule. Peñuela et al. (2020) assessed the forecast value for reservoir operations in the UK, focusing on maximising water supply and minimising pumping energy cost, and using the operations optimised against the deterministic worst-case scenario or ESP as a benchmark. They found that utilising ensemble forecasts can significantly enhance operational efficiency compared to relying on a deterministic scenario, whereas ESP is a hard-to-beat benchmark. Crippa et al. (2023) assessed the value of SFFs for agricultural water supply in Greece and found that SFFs can marginally improve reservoir operations with respect to using a simple reservoir operation rule. In quantifying SFFs value, they solely utilised the median of an ensemble of SFFs, hence leaving open the question of whether using the full ensemble and allowing for uncertainty in the optimisation process could bring more obvious advantages, as found in Peñuela et al. (2020).

This paper investigates the value of SFFs for informing reservoir operations in South Korea by assessing their potential to mitigate the impacts of three major historical drought events. We build on our previous works on the skill of seasonal precipitation and flow forecasts across catchments in South Korea, indicating that SFFs can be particularly suitable for predicting droughts. Specifically, we have compared the performance of precipitation forecasts from various forecasting centres and found that the European Centre for Mid-range Weather Forecasting (ECMWF) provides the most accurate forecasts in South Korea, and particularly during dry years (Lee et al., 2023a). Our subsequent research on translating seasonal weather forecasts into flow forecasts (Lee et al., 2023b), demonstrated that SFFs are generally more accurate than ESP up to 3 months ahead, and for even longer lead times in dry years.

In this study, we focus on two reservoir systems, Soyanggang-Chungju and Andong-Imha, which serve as crucial water sources for the country, including densely populated metropolitan areas such as the capital city, Seoul. To identify the optimal 'forecast-informed' reservoir operations over the selected historical drought events, we employ a multi-objective optimisation approach driven by SFFs. For comparison, we also consider ESP and two deterministic scenarios currently utilised by the national water agency in charge of reservoir operations (K-water). For each flow forecast/scenario, the value is assessed in terms of the chances of achieving better operational outcomes compared to historical operations. This approach can reveal the value in a more intuitive manner than using a synthetic benchmark, as done in most previous studies. To increase the robustness of our conclusions, simulation experiments are repeated with varying choices of forecast lead time, method for selecting a compromise solution between two conflicting objectives (minimising short-term supply deficit versus maximising the storage volume at the end of the hydrological year), and the temporal resolutions for repeating the multi-objective optimisation.



112   A key contribution of this paper is a methodology to quantify the forecast value and its sensitivity to key
113   experimental choices. We hope this will be useful for other studies and contribute to fostering more research into
114   the link between forecast accuracy and value in improving reservoir operations and mitigating drought damages.

## 2.   Study area and available data

### 2.1  Case study reservoirs and drought events

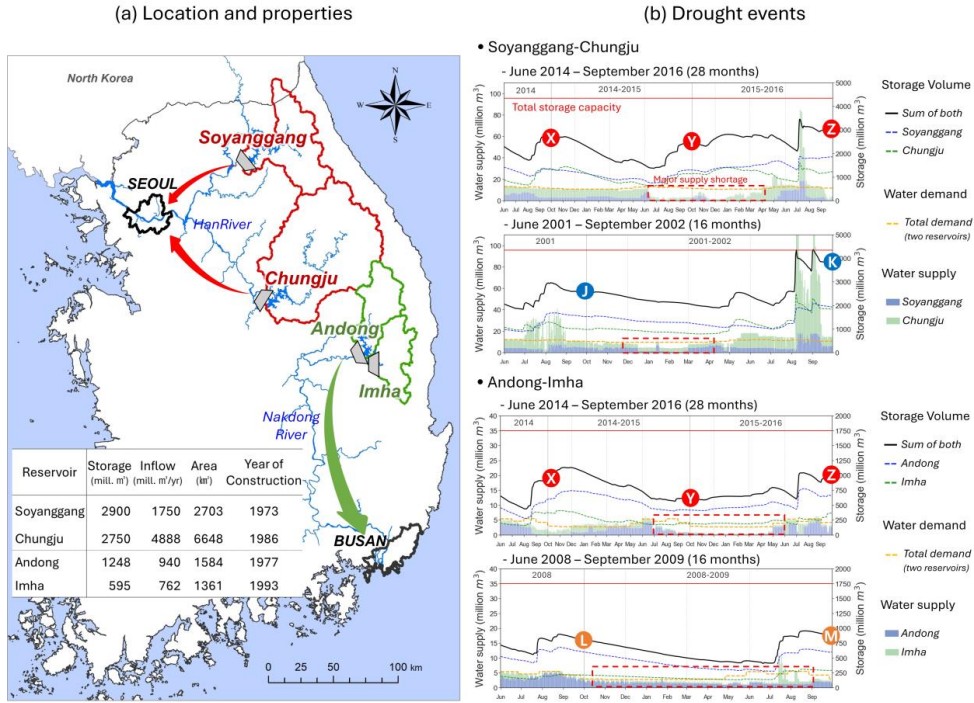

117

**Figure 1: (a) Location and properties of the studied reservoirs and their catchments. The green and red arrows represent the regions supplied by those reservoirs. (b) Daily reservoir operation records for the studied drought events (K-water, 2023). Points X, Y, Z, J, K, L, M represent the ends of the hydrological years (September 30th) which will be used as points for forecast value assessment.**

Currently, there are 20 multipurpose reservoirs in operation across South Korea, each playing a vital role in
national water resources management and the mitigation of water-related disasters (Park and Kim, 2014). This
study specifically focuses on two reservoir systems: Soyanggang-Chungju and Andong-Imha. As shown in Figure
1(a), Soyanggang and Chungju reservoirs are positioned upstream of the Han River, which runs through Seoul,
the capital city. They serve as primary water sources for the metropolitan area, including Seoul, with a population
of approximately 23 million people (K-water, 2023). In terms of total storage capacity, these two reservoirs also
stand as the two largest across the country. Andong and Imha reservoirs, which are in northernmost region of the
Nakdong River, supply water to plenty of cities alongside the river, including Busan, the second-largest city in
the country.
Both the Soyanggang-Chungju and the Andong-Imha reservoir systems are operated conjunctively by the national
water resources corporation, K-water, functioning effectively as a single reservoir. For instance, during periods
when one reservoir (e.g., Soyanggang) experiences reduced storage volume, the other reservoir (e.g., Chungju)
supplements the water supply. This inter-reservoir conjunctive operation is common and important in South Korea
for mitigating damages from droughts. Therefore, in this study, we treated both Soyanggang-Chungju and
Andong-Imha as integrated systems. A historical example illustrating the operational scheme for Soyanggang-
Chungju reservoir system, along with its conceptual framework, is provided in Figure S1 in the supplementary
material.





Figure 1(b) illustrates daily reservoir operation records during the historical drought events analysed in this study.
The recent catastrophic drought event, which persisted from 2014 to 2016, is included in the analysis for both
reservoir systems. This event caused severe damages such as regional water restrictions and nationwide crop
failures (K-water, 2018). During this period, the aggregated storage volume for Soyanggang-Chungju reached its
lowest in record (1373 million m³, 24.3%), and its third lowest record for Andong-Imha (434 million m³, 23.5%).
Additionally, we analysed the drought event from 2001 to 2002 for Soyanggang-Chungju, and the event from
2008 to 2009 for Andong-Imha.

**2.2 Observational data and seasonal weather forecasts**

This study utilises quality-controlled daily precipitation data from 49 in-situ stations distributed within the
catchments, as provided by K-water, along with daily temperature data from 37 in-situ stations managed by the
Korean Meteorological Administration (KMA). Unlike precipitation and temperature, potential
evapotranspiration (PET) data were computed based on the standardized Penman-Monteith method suggested by
the United Nations Food and Agriculture Organisation (Allen et al., 1998). We used the Thiessen polygon method
to calculate the mean areal data for each reservoir. For reservoir operation modelling, we used quality-controlled
daily reservoir operation records provided by K-water, including the storage volume, inflow, and water supply.
In generating theses records, K-water utilises a water balance equation, considering reservoir volume changes
from water level fluctuations and supplies. These reservoir inflow data are not corrected for removing the effect
of evaporation losses from the reservoirs.
For generating SFFs, we employed the seasonal weather forecasts provided by ECMWF (system 5). This choice
was based on our prior research, which demonstrated that generally ECMWF offers the most accurate precipitation
forecasts across South Korea (Lee et al., 2023a). ECMWF provides 25 ensemble forecasts from 1993 to 2016 and
51 ensembles since 2017 on a monthly basis, with a lead time extending up to 7 months ahead. To ensure
consistency with our previous works, we obtained ECMWF's seasonal weather forecasts datasets for precipitation,
temperature, and PET with a spatial resolution of 1×1°. We downloaded these data from the Copernicus Climate
Data Store and computed the monthly bias correction factors using datasets from 1993 to 2010.

**3. Methodology**

**3.1 Simulating forecasts-informed operations during a past drought event**

Figure 2 schematically outlines our methodology for simulating reservoir operations during a past drought event.
We began by compiling four distinct flow forecasts/scenarios using historical observational data and seasonal
weather forecasts from ECMWF system5 (Section 3.1.1). For each of this flow forecast/scenario, we conducted a
reservoir operation simulation experiment. In these experiments, we generated a set of Pareto optimal release
schedules, taking into account two conflicting objectives: securing storage volume and minimising supply deficit
(Section 3.1.2). A single compromise release schedule within this set is then selected using various Multi-Criteria
Decision-Making (MCDM) methods (Section 3.1.3). Following this, we simulated the evolution of the reservoir
systems until the next decision-making time step by feeding the chosen release schedule into a reservoir operation
model and using observed flow data. The aforementioned process was iteratively conducted until the end of the
simulation period. A more detailed description of our methodology and process is provided in Figure S2.





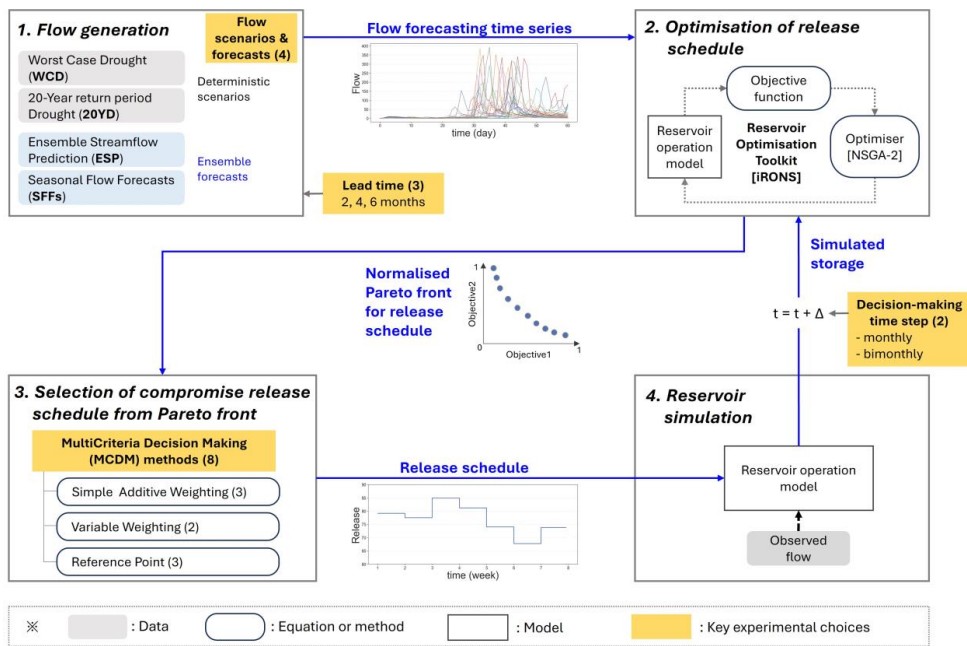


**Figure 2: Schematic diagram illustrating the reservoir simulation methodology employed in the study.**

### 3.1.1 Generation of deterministic flow scenarios and ensemble flow forecasts

In this study, we considered two deterministic scenarios, the Worst-Case Drought (WCD) and the 20-Year return period Drought (20YD), alongside two ensemble forecasts, ESP and SFFs. The WCD scenario was generated by comparing historical flow records and identifying the driest year for each reservoir. The 20YD scenario, currently employed in practical reservoir operations in South Korea, was obtained from K-water. To derive this scenario, K-water conducts a low-flow frequency analysis, a process similar to flood frequency analysis, utilising historical flow records spanning over 30 years (Ryoo et al., 2009; Jung et al., 2012).

To generate the ESP, we built an ensemble of 45 members for each weather variable (precipitation, temperature, and PET) based on historical observations from 1966 to 2010 and fed it into the conceptual Tank hydrological model (Sugawara et al., 1986, 1995) to generate a flow ensemble (Lee et al., 2023b). Lastly, we generated an ensemble of SFFs using ECMWF's seasonal weather forecasts (system 5) as input for the same hydrological model. Given the coarse spatial resolution ($1 \times 1°$) of the seasonal weather forecast data compared to the reservoir's catchment areas, we downscaled the data using the linear scaling method for bias correction. Further details regarding the structure, parameters, and performance of the Tank model, as well as the linear scaling method used for bias correction, are comprehensively documented in our previous paper (Lee et al., 2023b). For details regarding the flow forecast/scenario generation process, please refer to Figure S2.

To assess the accuracy of flow forecasts, we employed the Mean Error (ME) of monthly flow averaged across the entire simulation period. In calculating ME for ensemble forecasts, we considered their ensemble median. It is calculated as:

$$\text{Mean Error} = \frac{1}{N}\sum_{i=1}^{N}\left( Q_i^{Forecast} - Q_i^{Observation} \right) \qquad (1)$$

where, $N$ represents the total number of timesteps (months) in the simulation periods. $Q_i^{Forecast}$ and $Q_i^{Observation}$ are forecasted and observed monthly flow at time $i$ (month), respectively. When the ME is negative (positive), the forecast tends to underestimate (overestimate) the flow.

While ME is a simple measure of forecast accuracy, it does not account for the contributions of each member within the ensemble. Therefore, we also computed the forecast skill using the Continuous Ranked Probability Score (CRPS) and the Continuous Ranked Probability Skill Score (CRPSS), developed by Matheson and Winkler



(1976). While CRPS measures the absolute performance (score), CRPSS represents the relative performance (skill)
with respect to a benchmark, in our case the ESP. These metrics are computed as follows:
$$CRPS = \int [F(x) - H(x \geq y)]^2 \, dx \tag{2}$$
$$CRPSS = 1 - \frac{CRPS^{SFFs}}{CRPS^{ESP}} \tag{3}$$
where $F(x)$ represents the cumulative distribution of the SFFs ensemble, $x$ and $y$ are the forecasted and observed
flow. $H$ is called the 'Heaviside (or Indicator) function' and is equal to 1 when $x \geq y$ and 0 when $x < y$. $CRPS^{SFFs}$
and $CRPS^{ESP}$ are the CRPS of SFFs and ESP, respectively. When the CRPSS is positive, i.e., from 0 to 1, the
SFFs have skill with respect to ESP. Conversely, when the CRPSS is negative, ESP outperforms SFFs.
To exhibit the skill more intuitively, we employed the concept of 'overall skill', as introduced in our previous
research (2023a; 2023b). It represents the frequency with which SFFs outperform the benchmark (ESP) over a
specific period and can be expressed as:
$$Overall\ skill\ (\%) \ = \ \frac{\sum_{i=1}^{N}[\,H\,(CRPSS)\,(i)\,]}{N} \ \times \ 100\ (\%) \tag{4}$$
where $N$ is the total number of months in the simulation periods, the Heaviside function ($H$) is equal to 1 when
CRPSS ($i$) > 0 and 0 when CRPSS ($i$) ≤ 0. If the overall skill is greater than 55%, SFFs generally have skill with
respect to ESP across the period. However, if it is less than 45%, ESP outperforms SFFs. When the overall skill
is between 45% to 55%, both have an equivalent level of performance (Lee et al., 2023b).

### 3.1.2 Multi-objective optimisation of release schedule

Reservoir operations inherently involve managing multiple objectives often in conflict with each other (Zhou et
al., 2011; Vassoney et al., 2021). In terms of drought management, the amount of supply deficit shows an inverse
correlation with both the secured reservoir storage at the initial stage of the hydrological year (October 1st) and
the total inflow into the reservoir across the hydrological year (from October 1st to the subsequent September 30th).
In other words, inadequate storage at the outset of the hydrological year leads to substantial disruptions in water
supply, and the severity of these shortages further increases when the inflow is insufficient. This correlation is
supported by historical records, as illustrated in Figure S3.
To address this relationship in reservoir simulations, we established two operational objectives: the mean Squared
Supply Deficit (SSD, [million m³]²) over the drought event and the Storage Volume Difference (SVD, million
m³) relative to the reservoir's capacity by the end of the hydrological year. The rationale for squaring the supply
deficit is to incorporate risk hedging principles, aimed at strategically allocating water resources over time (You,
2013; Shiau, 2022). These two objectives are formulated as:
$$SSD = \frac{1}{T} \sum_{t=0}^{T} [\,Max(0, d(t) - Q(t))]^2 \tag{5}$$
$$SVD = \ Max(0, S_{max} - S) \tag{6}$$
where $T$ is the total number of weeks for which the flow forecast is available (i.e., $T$ equals the lead time in months
× 4), $d(t)$ and $Q(t)$ represent water demand and supply at time $t$ (week), respectively. $S_{max}$ is the storage capacity
of the reservoir (million m³), and $S$ is the simulated storage volume (million m³) at the end of the hydrological
year. When the end of hydrological year is not included in the simulation period, $S$ is set to the storage at the end
of the simulation. By definition, superior performance is associated with smaller objectives (SSD and SSD).
For the reservoir operation modelling and the optimisation of release schedules, this study utilises the 'interactive
Reservoir Operations Notebooks and Software' (iRONS) toolbox developed by Peñuela et al. (2021). This toolbox
offers a Python Jupyter Notebook based environment and encompasses a reservoir operation model based on a
mass balance equation. It also incorporates an optimiser that utilises the Non-dominated Sorting Genetic
Algorithm (NSGA-II) for the multi-objective optimisation of the weekly release schedules. Given that in multi-
objective optimisation problems, a single optimal solution that satisfies all objectives simultaneously is
unattainable (Lu et al., 2011; Malekmohammadi et al., 2011), NSGA-II identifies a set of non-dominated solutions
whose performance realise different Pareto optimal trade-offs between the two objectives. The performances
associated with these solutions visualised in the objective space constitute the so called "Pareto front" (Giagkiozis
and Fleming, 2014; Ni et al., 2022). We set the number of NSGA-II iterations for each Parto front generation to



one million. When optimising against ensemble forecasts, we generated a tentative Pareto front against each
ensemble member and then averaged them to identify a single optimal Pareto front based on the mean
performances of each solution.

### 3.1.3    Selection of a compromise solution from the Pareto front

A Pareto front, derived from Section 3.1.2, comprises one hundred non-dominated solutions (release schedules).
Therefore, a critical decision must be made to select a compromise release schedule from the Pareto front. The
methodology to select a single solution will be referred to as Multi-Criteria Decision-Making (MCDM) from now
on, as described in some of the previous literature (e.g., Wang and Rangaiah, 2017; Ni et al., 2022). MCDM
methods can provide helpful support to decision makers to select compromise alternatives for complex water
management issues (Afshar et al., 2011; Malekmohammadi et al., 2011; Zhu et al., 2017; Vassoney et al., 2021).
In this study, the MCDM method is employed as a way to mimic the selection that, in real world, would be made
by the reservoir operator when running forecasts through a reservoir operation optimisation model and being
returned a Pareto front. Given the significant uncertainty regarding how the operator would make this selection,
considering multiple MCDM methods provides a means to address this uncertainty in our assessment of forecast
value.
Various MCDM methods have been developed and utilised over the last several decades (Velasquez and Hester,
2013). Among them, this study employed eight distinct methods, which can be systematically categorized into
three groups: Simple Additive Weighting (SAW), variable weighting and reference point methods. Firstly, the
SAW method, which is frequently employed in decision-making (Arsyah et al., 2021), ranks the alternatives based
on their weighted sum performance (Fishburn, 1967). In this study, we consider the 'balanced' method where
equal weights are assigned to each objective, as well as the 'storage-prioritized' and 'supply-prioritized' methods,
which prioritize storage and supply, respectively.
Secondly, we propose the 'variable weighting' method, which reproduces more closely the thought process of
reservoir operators, who weight supply more when the storage is abundant and less when storage is scarce. We
applied this method in two ways: the 'simple selective' method, which adopts the same weights as in the SAW
methods but varying them depending on storage status, and the 'multi-weight' method, which applies more
detailed procedure to allocate weights based on storage status.
Lastly, the reference point method identifies the compromise solution on a Pareto front by measuring the distance
from a reference point. In this study, we applied three approaches: the 'utopian point', 'knee point', and 'TOPSIS'
methods. The utopian point method selects the solution on the Pareto front that minimises the Euclidean distance
from the utopian (or ideal) point, which represents the theoretical perfect solution (Lu et al., 2011). The knee point
method selects the knee point, which is a point where the curvature of the Pareto front is maximum (Das, 1999).
Among various methods for detecting the knee point, we employed the Minimum Manhattan Distance method
which is known for its simplicity and robustness (Chiu et al., 2016; Li et al., 2020). The TOPSIS method selects
a point with the shortest Euclidian distance from the ideal point and the longest distance from the anti-ideal point
as the compromise solution (Hwang and Yoon, 1981; Liu, 2009). This is a widely chosen method (Tzeng and
Huang, 2011; Wang and Rangaiah, 2017) including the United Nations Environmental Program (Chen, 2000; Zhu
et al., 2015).
Detailed information on the MCDM method and normalisation of a Pareto front, including equations, merits, and
demerits, is provided in the supplementary material (Section S1 and S2).

## 3.2  Measuring the forecast value and its sensitivity to experimental choices

As shown in Figure 3, for each drought event, the reservoir simulation routine described in Figure 2 is repeated
for each forecast/scenario and with various combinations of key experimental choices. These choices include the
forecast lead time, the MCDM method, and the decision-making time step. Therefore, the total number of
experiments for each drought event amounts to 192 (3 lead times × 8 MCDM methods × 2 decision-making time
steps × 4 flow forecasts/scenarios).



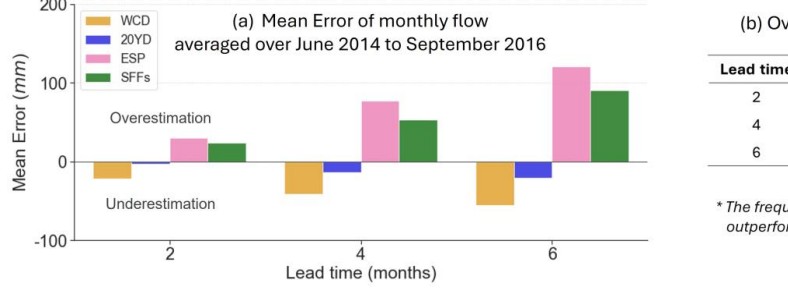

**296**

**297** **Figure 3: Key experimental choices and their combinations for reservoir operation simulations. Each simulation**
**298** **experiment (Exp.1 to 192) is conducted according to Figure 2.**

**299** For each experiment and forecast product, we computed two performance indicators (objectives), representing
**300** supply deficit (SSD) and storage volume (SVD) as in Eqs. 5 and 6 but using the simulated storage and release
**301** time series (i.e., coming from step 4 in Figure 2). We then calculated the same indicators using the observed
**302** storage and release, so to quantify the performance of the historical operations, which we use as a benchmark.
**303** Unlike previous studies (e.g., Turner et al., 2017; Peñuela et al., 2020; Crippa et al., 2023) that analysed
**304** improvements in performance indicators separately, here we propose a new and simply way to take into account
**305** the improvement in both indicators simultaneously. In fact, performance indicators generally exhibit a trade-off
**306** relationship with each other, so that an improvement with respect to the benchmark for one indicator may come
**307** at the price of a loss in the other. Analysing them independently from one another obfuscates these trade-offs.
**308** To overcome this issue, here we calculated the difference in each indicator (simulated - historical) in each
**309** experiment and defined the forecast value as the number of experiments where this difference is negative for both
**310** indicators. In fact, since we aim to minimise both indicators, negative differences in both indicate that the
**311** simulated operations outperform the historical operation. This method provides an intuitive and practical
**312** understanding of forecast value, as it directly relates to the chances of achieving better operational outcomes
**313** compared to historical operation, taking into account operational trade-offs and factoring in the uncertainty in the
**314** key experimental set-up choices. Last, we analysed the sensitivity of forecast value to those key experimental
**315** choices. This analysis serves as a useful tool in pinpointing the primary determinant of forecast value and offering
**316** insights for optimising setup choices to maximise the value for drought management.

**317** **4. Results**

**318** For clarity of illustration, in Section 4.1, we first present the results for one event and reservoir system: the drought
**319** that occurred in Soyanggang-Chungju from 2014 to 2016 (Figure 1(b)). In Section 4.2, we expand our results to
**320** include other reservoir systems and events, aiming to draw a more comprehensive conclusion on the value of SFFs
**321** and its key controls.

**322** **4.1 Simulation results for the 2014-2016 drought in Soyanggang-Chungju reservoirs**

**323** **4.1.1    Accuracy and skill of seasonal flow forecasts**

**324**



**Figure 4: (a) Mean Error of monthly flow (simulated – observed) for Soyanggang-Chungju reservoir system averaged**
**from June 2014 to September 2016 for 2-month lead time. (b) The overall skill, which represents the frequency of SFFs**
**outperforming ESP across the simulation period.**
Figure 4(a) illustrates the Mean Error of monthly flow (see Eq. 1) for lead times of 2, 4, 6 months and different
type of flow forecast/scenario (WCD: yellow, 20YD: blue, ESP: pink, SFFs: green). As shown in the figure,
deterministic scenarios (WCD and 20YD) exhibit smaller errors compared to the ensemble forecasts. This is not
surprising, as the WCD and 20YD scenarios are designed to mimic dry conditions and we are now evaluating
accuracy on a severe drought event. Ensemble forecasts, particularly, show a systematic bias towards
overestimating flows, with this tendency being more pronounced in ESP compared to SFFs. This pattern is
consistently observed across different reservoir systems and events, as further illustrated in Figure S4.
Figure 4(b) shows the overall skill (see Eq. 4), indicating the frequency with which SFFs outperforming ESP
across the simulation period. In this specific event, the overall skill exceeds 60%, indicating that SFFs perform
better compared to ESP. However, our results applied to different reservoir systems and events, suggest that SFFs
do not consistently outperform ESP, and the overall skill typically decreases as lead time increases (see Figure
S4).
**4.1.2    Reservoir simulations and their performances**
The simulated reservoir operation results are illustrated in Figure 5, showing the storage volume (a) and
cumulative squared supply deficit (b) generated using WCD (yellow), 20YD (blue), ESP (pink) and SFFs (green).
For each flow forecast/scenario, there are 48 simulation outcomes resulting from different combinations of the
experimental choices (3 lead times × 8 MCDM methods × 2 decision-making time steps). Higher storage volume
compared to historical operation (black line) is preferable and vice versa for cumulative squared supply deficit.
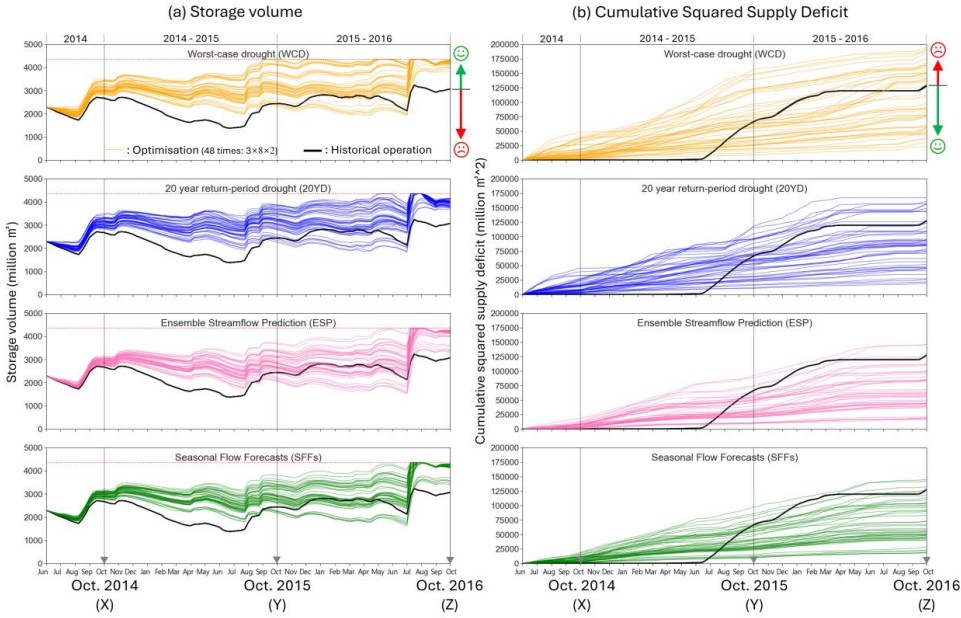
**Figure 5: Simulated reservoir operation results for Soyanggang-Chungju from June 2014 to September 2016 in terms**
**of (a) storage volume and (b) cumulative squared supply deficit. From top to bottom, the rows represent simulation**
**results generated by using WCD (orange), 20YD (blue), ESP (pink), and SFFs (green), respectively. Each sub-figure**
**has 48 simulated results (coloured lines, 3 lead times × 8 MCDM methods × 2 decision-making time steps) and a single**
**historical operation (black line).**
Figure 5(a) demonstrates that, in most cases, reservoir simulations achieve higher storage volumes throughout the
event in comparison to historical operation. By the end of the simulation period (September 30th, 2016, Z), all




forecast-informed operations effectively replenished the reservoir system more than the historical operation,
particularly due to the wet event observed towards the end of the simulation (around July 2016). Deterministic
scenarios (upper two rows) offer slightly superior results for securing storage volume compared to ensemble
forecasts (lower two rows). Conversely, as depicted in Figure 5(b), reservoir simulations using deterministic
scenarios tend to exhibit more supply deficits than ensemble forecasts. This trend arises from the underestimation
of flow by deterministic scenarios (see Figure 4(a)), which results in reduced water releases and consequently
increased supply deficits.
In summary, forecast-informed operations using deterministic scenarios generally do better at securing storage
volume while using ensemble forecasts is better for minimising supply deficit. This result is further corroborated
by our additional analysis across different events and reservoir systems, as depicted in Figure S5.

### 4.1.3 Value of seasonal flow forecasts

Figure 6 depicts the differences in achieved performance indicators (SSD and SVD) between simulated operations
and historical operation at distinct time points X, Y, and Z (see Figure 5), corresponding to the end of hydrological
years (September 30th). Coloured circles in the figure denote the type of flow forecast/scenario used in simulations
(following the same colour code as in Figure 5), and there are 48 circles (3×8×2) in each colour, corresponding to
the combinations of 3 lead times, 8 MCDM methods, and 2 decision-making time steps. Circles positioned below
(above) zero for both the x and y axes, i.e., within the green (red) shaded area, indicate experiments where reservoir
simulations achieve better (worse) performance compared to historical operation. The count of circles within the
green shaded area (bottom-left quadrant) represents forecast value, indicating the chances of simulated reservoir
operations outperforming historical operation, as detailed in Section 3.2.

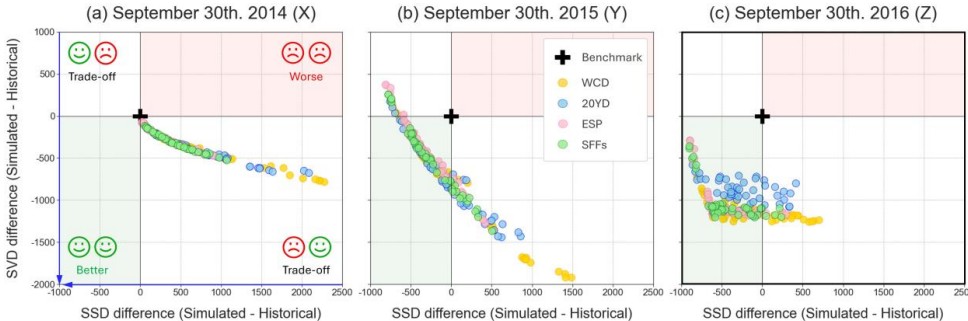

**Figure 6: Difference of SSD (x-axis) and SVD (y-axis) between historical operation (black cross) and simulated operations using different flow forecasts/scenarios (coloured circles) in Soyanggang-Chungju during the 2014-2016 drought. Performances are calculated on September 30th in (a) 2014, (b) 2015 and (c) 2016. Each sub-figure shows 48 points for each flow forecast/scenario (WCD, 20YD, ESP, SFFs), resulting from different combinations of key experimental choices (3 lead times × 8 MCDM methods × 2 decision-making time steps).**

At the initial stage of simulation, as shown in Figure 6(a), simulated forecast-informed operations only exhibit a
trade-off relationship with historical operation. All circles are distributed in the bottom-right quadrant, indicating
that the historical operation prioritized water supply over storage volume until the end of September 2014 (X).
However, as the impact of forecast-informed operations accumulates (i.e., the period of simulation moves from X
to Z), more circles tend to fall in the green shaded area where simulated operations outperform historical operation.
Specifically, as shown in Figure 6(c), the majority of simulations not falling within the green shaded area by the
end of the simulation (September 30th, 2016 (Z)), are associated with deterministic scenarios (yellow and blue
circles). These findings are consistently demonstrated with our experiments applied to other drought events and
reservoir systems, as presented in Figure S6.

### 4.1.4 Sensitivity of forecast value to key experimental choices

Figure 7 presents the sensitivity of forecast value to the choice of forecast lead time (a), MCDM method (b),
decision-making time step (c) and type of flow forecast/scenario (d) at the end of simulation period (September
30th, 2016, corresponding to Figure 6(c)). The maximum number on the y-axis in each sub-figure represents the
total number of simulation experiments conducted for a particular experimental set-up choice. For example, in
Figure 7(a), the lead time is fixed at 2, 4 or 6 months (horizontal axis) and for each of these choices there are 64
experiments (see range of vertical axis), resulting from the combination of 8 MCDM methods, 2 decision-making



time steps, and 4 flow forecasts/scenarios. When an experimental choice (x-axis) correlates with a higher forecast value (y-axis), it indicates that using that specific experimental choice can lead to greater operational benefits for managing droughts.

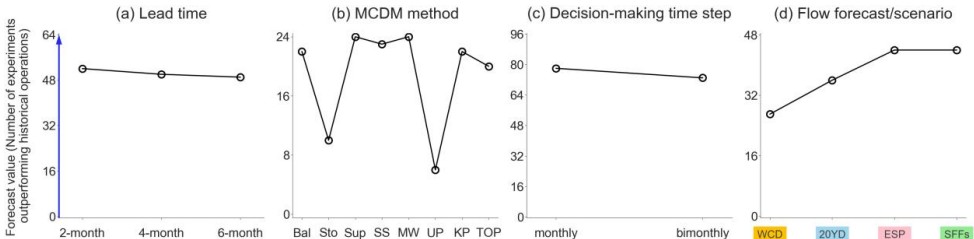

**Figure 7: Forecast value (y-axis) against key experimental choices: (a) forecast lead time, (b) MCDM method, (c) decision-making time step and (d) type of flow forecast/scenario for Soyanggang-Chungju reservoir system on September 30th, 2016. (Refer to Figure 3 for the acronyms corresponding to the MCDM methods.)**

As shown in Figure 7(b), forecast value varies significantly depending on the MCDM method used for selecting a compromise solution from the Pareto front, suggesting that this choice is a key control of forecast value. In this specific drought event, for example, using the storage-prioritized (Sto) and utopian point (UP) methods leads to a much lower forecasts value compared to using the other methods. Conversely, the choice of lead time (Figure 7(a)) and decision-making time step (Figure 7(c)) has a marginal influence.

Importantly, Figure 7(d) demonstrates that the value can also be influenced by the type of flow forecast/scenario. It highlights that a higher value is attained using ensemble forecasts (ESP, SFFs) than deterministic scenarios (WCD, 20YD). In this particular case, there is no difference in forecast value between ESP and SFFs.

## 4.2 Simulation results for other reservoir systems and drought events

### 4.2.1 Sensitivity of forecast value to key experimental choices

Having analysed the forecast value and its key controls for one drought event in one reservoir system, Figure 8 illustrates whether similar or contrasting results are found in the other three events and reservoir systems considered in this study (see Figure 1(b) for a description of these events). Note that this figure incorporates the result from Soyanggang-Chungju for the 2014-2016 drought event shown in Figure 7 (white circles connected by solid line).

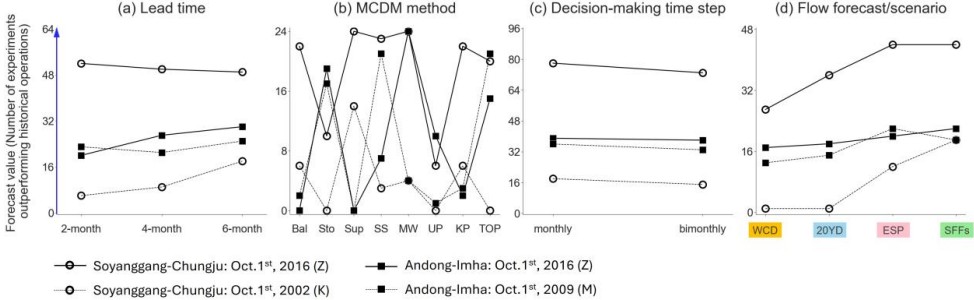

**Figure 8: Forecast value (y-axis) against key experimental choices including (a) lead time, (b) MCDM method, (c) decision-making time step and (d) type of flow forecast/scenario for Soyanggang-Chungju (○) and Andong-Imha (■) at the end of different drought events (points Z, K, M in Figure 1(b)).**

Figure 8(b) confirms the substantial influence of the choice of MCDM method on forecast value. However, it also highlights the significant variability with events and reservoir systems. For example, the multi-weight method (MW) leads to a high value during the 2014-2016 drought event for both reservoir systems but a low value for the other events. Or the TOPSIS method (TOP) generally provides relatively higher forecast value for three events but a notably low value during the 2001-2002 drought in Soyanggang-Chungju.





The higher value of ensemble forecasts (ESP and SFFs) is also confirmed in Figure 8(d), particularly in
Soyanggang-Chungju reservoir system, whereas their advantage over deterministic scenarios (WCD and 20YD)
is less pronounced in Andong-Imha. Lastly, Figure 8(a,c) indicate that increasing the forecast lead time or
decreasing the decision-making time step slightly improves forecast value. Yet again, this improvement appears
relatively marginal when compared to the impact of the chosen MCDM method or flow forecast/scenario.

### 4.2.2    Relationship between forecast accuracy and value

Figure 9 illustrates the overall relationship between the accuracy of each flow forecast/scenario and its value in
informing decision-making for enhanced drought management. For this figure, we only used experiments with a
6-month lead time and monthly decision-making, to closely mimic current reservoir operations practices in South
Korea. Note that excluding other options for these two experimental choices should not undermine the robustness
of our conclusions as the sensitivity analyses in previous sections have shown that these choices have low impact
on forecast value.

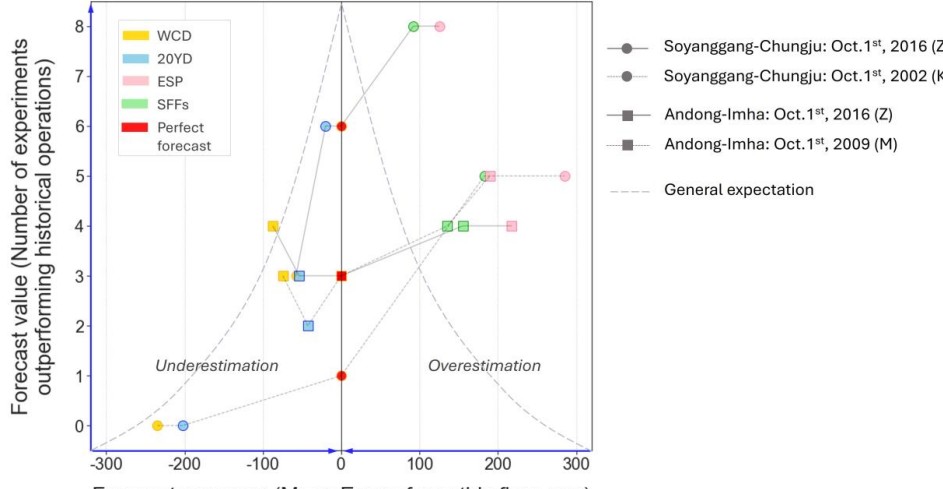


**Figure 9: Relationship between forecast accuracy (Mean Error of monthly flow, mm) and value at the end of the
simulation period for different drought events (2002, 2009 and 2016) at Soyanggang-Chungju and Andong-Imha
reservoir systems. For each event and system, the figure shows five points corresponding to simulated forecast-
informed operations using different forecasts/scenarios (orange: WCD, blue: 20YD, pink: ESP, green: SFFs, red:
perfect forecast). The direction of the blue arrows indicates higher performance (high value, low error). The grey
dashed lines represent the general expectation on the relationship between forecast accuracy and value.**

Figure 9 demonstrates that despite the higher accuracy of deterministic scenarios, as evidenced by general
proximity of yellow and blue points to zero on the x-axis, ensemble forecasts (pink and green points) are expected
to result in a higher value. These findings deviate somewhat from the general expectation on the relationship
between forecast accuracy and value (represented by the dashed grey lines) that higher accuracy would lead to
higher value. When comparing the accuracy between ensemble forecasts, SFFs demonstrate a slight advantage
over ESP, with a tendency for smaller overestimations. In terms of forecast value, however, there are no significant
differences between them, indicating the operational benefits obtained from using SFFs and ESP appear to be
comparable.
Figure 9 also includes the value obtained from optimizing operations against the perfect forecast scenario, depicted
by red symbols, that is, using observations of future flows as if they were 'perfect' forecast (note that, by
construction, this scenario is associated with zero error on the horizontal axis of Figure 9). Surprisingly, this figure
indicates that the value of perfect forecast in our experiments is lower than that of ESP and SFFs. This
counterintuitive result stems from the fact that even with perfect knowledge of flows within the optimisation
horizon (i.e., the forecast lead time), perfect forecast does not resolve the uncertainty about future flows beyond
that horizon. Therefore, acknowledging uncertainty during the optimisation horizon, as done when using ensemble





forecasts, yields more cautious operations that in the long-term prove to be more robust against adverse events
not seen during the optimisation.
**5. Discussion**
**5.1 Value of SFFs in informing decision-making for managing droughts**
Our findings highlight the higher value of ensemble forecasts over deterministic scenarios, aligning with several
previous studies. For example, Peñuela et al. (2020) demonstrated that employing ensemble forecasts can yield
higher operational benefits compared to using deterministic (worst-case) scenario in a water supply two-reservoirs
system in the UK. The higher value of ensemble forecasts for informing flood control decisions was also
demonstrated by Fan et al. (2016). They compared the value using the ensemble mean versus using the full SFFs
ensemble and found that the latter notably enhanced forecast value. However, our research also revealed that the
extent to which ensemble forecasts yield higher value can vary significantly depending on the reservoir systems,
as the enhancement of operational benefits was more evident in the Soyanggang-Chungju than in the Andong-
Imha reservoir system.
While our findings emphasize the importance of considering forecast uncertainty when optimising reservoir
operations, no significant difference in forecast value was found between the two ensemble forecasts. This is
consistent with the findings of Peñuela et al. (2020), who similarly observed no notable difference in the value of
ESP and SFFs. Given the lower computational cost and higher practical experiences of generating ESP, the latter
remains a hard-to-beat reference.
This study includes a sensitivity analysis that examines how forecast lead time, MCDM method, decision-making
time step and type of flow forecast/scenario affect value. Although we found some improvements in forecast value
with longer lead times, their impact was generally marginal. A prior study by Yang et al. (2021) also evaluated
the influence of lead time, ranging from 10 to 30 days, on forecast value for hydropower and water supply. They
argued that considering a longer lead time for forecast-informed operations may enhance the value. However, the
lead times they examined were considerably shorter than those in our study, which makes direct comparisons with
our study challenging.
Our findings show that the MCDM method used to select a compromise solution from the Pareto front is a key
determinant of forecast value, but no single MCDM method that consistently performs best across different events
and reservoir systems was identified in this study. Previous research also has demonstrated that different MCDM
methods have distinct performance characteristics (Velasquez and Hester, 2013; Taherdoost and Madanchian,
2023). The highly variable performance of MCDM methods depending on reservoir systems and drought events
emphasizes the significance of using ensemble forecasts in reservoir operations, as they consistently bring
operational benefits.
While identifying an optimal MCDM method was not possible in this study, practical guidelines can be offered
for applying each method based on their inherent characteristics. Firstly, the SAW method is straightforward to
apply and may be particularly advantageous for reservoirs with obvious operational purposes or characteristics.
Specifically, the supply-prioritized method might be well-suited for a reservoir with ample storage capacity but
lower demand, while the storage-prioritized method would be useful for reservoirs with a high risk of causing
significant economic or social damages when facing a substantial supply deficit. Secondly, the performance of
the variable weighting method can be highly dependent on subjective choices in determining the appropriate
weights and storage ranges. Therefore, sufficient operational records are essential for effectively applying this
method. Conversely, the reference point method, offering a geometric estimation of the compromise solution, may
prove advantageous for reservoirs with limited operational history.
We also found that using ensemble forecasts, despite their limited accuracy, yields more benefits than using
deterministic scenarios, even the perfect forecast. The lower performance of the perfect forecast scenario can be
attributed to its finite lead time and its reliance on a single flow scenario without accounting for uncertainty. Few
previous studies also reported that forecast-informed operations forced by ensemble forecasts often deliver
comparable or higher performance compared to the perfect forecast scenario (Zhao et al., 2011; Fan et al., 2016;
Ficchì et al., 2016).
Bias correction of seasonal weather forecasts, such as precipitation, is a widely addressed issue concerning the
performance of SFFs (Shrestha et al., 2017). In this study, we utilised bias corrected SFFs, building on our
previous findings that demonstrated the effectiveness of bias correction in improving the accuracy of SFFs (Lee
at al., 2023b). While the positive impact of bias correction on SFFs is widely documented in the literature (e.g.,
Lucatero et al., 2018; Tian et al., 2018; Pechlivanidis et al., 2020), a previous study noted that bias correction may
potentially reduce performance under extreme conditions (Crochemore et al., 2016). Our supplementary



experiment, presented in Figure S7, investigates the influence of bias correction on forecast value. The result indicates that bias corrected SFFs generally yield higher value compared to SFFs without bias correction. However, to fully validate the impact of bias correction on the value, further research applying our methodology across diverse reservoirs and drought events is necessary.

### 5.2 Limitations and directions for future research

This study provides valuable insights into the value of SFFs in informing decision-making for managing droughts; however, it is essential to acknowledge several limitations. Firstly, in assessing forecast value, we proposed a methodology that uses historical operational result as a benchmark. While it offers more intuitive value comparison, it is important to recognize that this benchmark may be influenced by complex decisions considering various internal and external circumstances. For example, decisions on water releases may be adjusted based on additional water supplies from external sources such as neighbouring reservoirs or rivers.

Secondly, while we assessed forecast value across two reservoir systems and historical drought events, these case studies may not be sufficient to draw general conclusions. Since our results have demonstrated the dependency of forecast value on reservoir systems and events, it is crucial to continue further application efforts to establish more general patterns in the relationship between accuracy and value, as well as to compare the performance between different forecast products.

In this study, evaporation from a reservoir is not considered. In South Korea, direct measurements of reservoir evaporation are rarely conducted, which poses challenges to ensuring the reliability of indirect evaporation estimation. Recent research by Park et al. (2024) introduced an empirical formula to estimate reservoir evaporation specifically for Yongdam Reservoir, which is uniquely equipped with direct evaporation measurements. The study highlighted the importance of validating this formula for its applicability to other reservoirs. However, reservoir evaporation tends to intensify during extreme droughts, resulting in increased loss of storage volume (Wurbs and Ayala, 2014; Shah et al., 2024). Thus, further studies incorporating reservoir evaporation based on reliable estimating formulas are necessary.

Our analysis focuses on two operational objectives: securing storage and minimising water deficit, to centre our attention on reservoir operations. However, there are various other objectives worth considering for simulating reservoir operations, such as potential economic damages from droughts or benefits of risk hedging. Although quantifying those objectives is challenging, incorporating them into a multi-objective approach for drought management could significantly assist water managers.

### 6. Conclusions

This study demonstrates both the potential usefulness and the limitations of SFFs in informing reservoir operations for managing droughts. While deterministic scenarios (WCD, 20YD) exhibited higher accuracy, the value achieved from using ensemble forecasts (ESP, SFFs) was higher. This result emphasizes the significance of considering flow forecast uncertainty when optimising reservoir operations and demonstrates that higher forecast accuracy does not necessarily translates into higher value. Our study also suggests that forecast-informed operations using ensemble forecasts can reduce supply deficit and increase storage conservation compared to historical operations during past drought events. However, no evidence was found supporting that SFFs can lead to greater value over conventional ESP at present. As seasonal weather and flow forecasting technology continuously evolves and improves, this conclusion is provisional. Therefore, it will be important to continuously evaluate the performance of SFFs in enhancing reservoir operations as new forecasting products become available. Our sensitivity analysis also shows that the MCDM method used to select a compromise release scheduling from a Pareto front is a key control of forecast value. This suggests that the operator's prioritisation of competing objectives is crucial in determining forecast value.

Beyond our specific results, this study also proposed a new simple method to assess the forecast value that simultaneously takes into account the trade-offs between operational objectives and the uncertainty stemming from key set-up choices for the simulation experiments. This is achieved by counting the number of simulation experiments that outperform benchmark operations (the historical operations in our case) simultaneously for two objectives. This straightforward performance metric may be useful for quantifying forecast value in a practical and intuitive manner across a wide range of studies for water resources management, beyond drought management, such as hydropower, flood control, and others.

Considering the ongoing escalation in the frequency and severity of drought events worldwide, including South Korea, our effort is both timely and essential for finding novel solutions to mitigate future drought damages. We hope that our findings, methodology and framework stimulate further research to deepen our understanding and expand the practical application of SFFs in reservoir operations for enhancing drought management.



*Code and data availability*. The iRONS package used for reservoir operation modelling and optimisation is
available at https://ironstoolbox.github.io/. In manipulating and simulating seasonal weather and flow forecasts,
the SEAFORM (SEAsonal FORecast Management) and SEAFLOW (SEAsonal FLOW forecasts) Python
packages can be useful and are available at https://github.com/uobwatergroup/seaform, and
https://github.com/uobwatergroup/seaflow, respectively. ECMWF's data are available under a range of licences.
Reservoir and flow data are made available by the K-water and can be downloaded from https://www.water.or.kr/.
*Author contributions*. YL designed the experiments, with suggestions from co-authors. YL developed the
workflow and performed simulation. FP and MARR participated in the discussions on the interpretations of
results. AP provided YL with modelling technical support. All authors reviewed and contributed to the writing of
the manuscript.
*Competing interests*. The authors declare that they have no conflict of interest.
*Acknowledgements.* Yongshin Lee is funded through a PhD scholarship by K-water (Korea Water Resources
Corporation). Francesca Pianosi is partially funded by the Engineering and Physical Sciences Research Council
(EPSRC) "Living with Environmental Uncertainty" Fellowship (EP/R007330/1). Andres Peñuela is funded by
the European Research Executive Agency (REA) under the HORIZON-MSCA-2021-PF-01 grant agreement
584    101062258.

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
