# Peer review of "Value of seasonal flow forecasts for enhancing reservoir operation and drought management in South Korea"

_EGUsphere, 2024_

## Author Comment (AC1)

**Reply To Referee#1 Comments**

We thank referee for the valuable comments. Our replies to all comments are shown in blue and the original referee's comments are shown in black.

In this manuscript, the authors present a comparison between various ensemble forecasts and deterministic scenarios for informing reservoir operations for two reservoir systems and three drought events in South Korea. The evaluate gains in terms of the forecast accuracy and skill, and in terms of the operational value related to storage and supply. They additionally test the sensitivity of the results to methodological choices, decisions usually implicitly made in research studies. The research questions are interesting and the manuscript is overall clear and concise. Below are a few comments that I hope will be helpful to revise the manuscript for publication.

Thank you for your valuable and insightful comments to our paper. We are committed to address them in our revision.

**General points:**

-A clarification of the methods is needed, especially regarding the lead times and the decision-making time steps. For example, how can a decision be made every two months (i.e., bimonthly) for a forecast with two months lead time? Including a graphic illustrating the timeline between the forecast generation and the last decision made for a single water year would be really helpful, I think.

→ Thank you for your comment. We will revise the text to improve clarity and, since it is difficult to explain with text alone, we will also include a conceptual figure, as illustrated below, in the supplementary material to provide more detailed insights into the methodologies.

[Figure]

**Figure R1. Conceptual examples of our continuous reservoir simulations (2014-2016) with various experimental choices.**

-More reflection is needed on the plausible physical explanations for some of the results to add some depth. For example, see my comment on L427-429.

→ We agree with you and are aware that this is very important point. However, it was challenging to derive physical explanations from our results confined to 2 reservoir systems and three drought events. This limitation is discussed in 5.2 'limitations and directions for future research'. For further details, please refer to the answer for L427-429 on page 11.

-Please discuss the shortcomings associated with evaluating only two attributes of the forecast performance (i.e., accuracy and skill). Calculating more attributes, like correlation, variance and reliability, would give a fuller picture, which could impact the conclusion you draw on L448-450 regarding the link between forecast performance and value.

→ We agree with you and we will add a sentence in the Discussion to raise this point, for example saying: "In this study we only evaluated two attributes of the forecast performance (accuracy and skill) but other attributes may also be considered, such as correlation, variance and reliability, which could return different outcomes in terms of the comparison between forecasts products and therefore the relationship between performance (i.e. the level of agreement between forecasts and observations) and value (the usefulness to inform decisions).

-Are the codes you developed for the evaluation shared anywhere for others to follow your approach more easily? If not, please consider making them available.

→ We are currently organizing the code developed in this study for evaluation purposes and plan to make it publicly available in the iRONS python package (https://ironstoolbox.github.io/). We will add the link to access our code in the final version of the manuscript.

**Specific points:**

-L23: In the abstract, please specify what key choices you're looking at.

→ Yes, we will specify the types of key choices in the abstract.

-L113-114: I would move this last sentence to the conclusions instead as it seems a bit out of place in the introduction.

→ We agree with you and will remove the last sentence from the introduction.

-L122-130: Could you give a brief description of the hydrological regime of both regions? E.g., When are the peak flows? What drives runoff generation?

→ They share similar hydrological scheme with peak flows in July or August due to monsoon or typhoon. We will include a brief description of the hydrological regimes in the manuscript.

-L140: On L39 the dates for this event are 2013-2016. Please clarify.

→ In national perspective, 2013-2017 is regarded as drought event (Line-39, K-water, 2018). However, for the two studied reservoir systems, total inflow in 2013 was above average and the drought event lasted until 2016 (Line 140). To avoid confusion, we will unify the drought period as 2014-2016 in the manuscript, aligning it with our simulation period.

-L159-162: What is the initialization frequency of the forecasts and what time period is covered by the forecasts you generated for this study?

→ Forecasts are initialised every month and the time period covered by the forecasts are the same as studied drought events (2001-2002, 2008-2009, 2014-2016). We will add details in the manuscript.

-L162-163: The time period based on which the correction factors are calculated overlaps with the drought events in 2001-2002 and 2008-2009. This could be an unfair advantage for these events. Please clarify. Same comment for the ESP generation explained on L186.

→ Thank you for this comment. We used the time period 1993-2010 for generating the bias correction factors because of concerns about data sufficiency. For example, when analysing drought events from 2001 to 2002, only 7 years of data from 1993 to 2000 would be available if we tried to avoid overlap with the event period. We agree that incorporating overlapped years could potentially provide unfair advantages, but using an insufficient amount of data for generating bias correction factors can also lead to significant issues. Johnson and Sharma (2012) and Maraun et al. (2010) suggest that larger datasets help ensure more accurate bias corrections by capturing the variability of the data better and reducing the influence of outliers. By fixing the time period from 1993 to 2010, we ensure a more reliable and robust calculation. We have applied the same time period constraint to the ESP calculation until 2010 to maintain consistency of our study. We will clarify this in the manuscript.

-L167: Can you briefly list the four different forecasts/scenarios here as well?

→ Yes, we will add brief list of the forecasts and scenarios here.

-L172-173: What is the temporal aggregation of the forecasts/scenarios, based on which the decisions are made? E.g., Weekly, monthly, etc.

→ All forecasts and scenarios are initially generated at a daily time scale, but in optimising and simulating reservoirs, we aggregated this data to a weekly time scale. Decisions were made every month (monthly) or every two month (bimonthly) mimicking the practical reservoir operations. We will clarify this temporal aggregation detail in the manuscript.

-L173: Please specify how much time there is in between each decision.

→ We will specify decision making time steps which are 1 and 2 months.

-L174: Is the process iteratively conducted at the start of each month? The frequency is unclear.

→ This reservoir optimisation and simulation process is iteratively conducted every month (for monthly decision-making) or every two months (for bimonthly decision-making) throughout the simulation period (e.g. Jun. 2014 to Sep. 2016) (refer to Figure R1). We will clarify this in the sentence.

-Figure 2: Very nice graphic!

→ Thank you.

->You could refer the readers to each figure compartment being described at the start of each sub-section below.

→ Yes, it would be a good idea. We will add this information in the manuscript.

->Could you specify whether bimonthly refers to twice a month or every two months please?

→It means every two months as illustrated in Figure R1, and we will clarify this in the text and the Figure 2.

->It would be useful to add a short section (3.1.4) for the reservoir simulation step, both to have coherence between the numbering of the sections and the boxes in the figure (i.e., step 1 is explained in 3.1.1, step 2 in 3.1.2, etc.) and to provide some information on how this is done (e.g., I don't quite understand how decisions are made at different time steps with forecasts that cover different lead times and how often a new forecast is produced).

→We thank you for this advice and will add section 3.1.4 for the reservoir simulation step. In this section, we will include a reference to the Figure R1 (shown above).

-L178-193: Please provide more information about the forecasts' generation, with regards to the: simulation periods, forecast time steps, initialization dates, lead time (please also explain how you define lead time with a concrete example as different research groups define them differently, e.g., lead month 0 vs. 1), ensemble size for the SFF.

→ Flow scenarios and forecasts have a daily time step; the simulation periods are those specified in Figure 1. We generated and applied lead times of 2, 4, and 6 months from current time step (i.e. 2 months of lead time corresponds to the next 60 days). The SFF ensemble includes 25 members until 2016, and 51 members since 2017. We will add these details in the manuscript.

-L181-182: Operationally, with what lead times and at what time steps are the decisions made currently by K-water? This would help contextualize your methodological decisions.

→ K-water revises their weekly release schedule every month, utilizing a 20-year return period drought scenario for the upcoming 3 to 6 months. We will include this information in the manuscript.

-L185-189: Could you comment on the difference in ensemble sizes between the ESP and the SFF and the potential impacts on the performance evaluation?

→ Followed by your comment (L178-193), the ensemble sizes of ESP (45) and SFFs (25 until 2016, 51 since 2017) will be added in the manuscript. However, evaluating the impact of the ensemble size is out of the scope of this study given the limited number of drought events.

-L186-187: Could you give a bit more information about the Tank hydrological model, such as its spatial resolution, how it was calibrated, how the initial conditions were obtained, and what its performance in simulation is for the basins considered here.

→ We acknowledge the importance of this information and will add information about the model to this manuscript. However, we will keep this brief since our previous paper published in Hydrology and Earth System Sciences (Lee et al., 2024) provides comprehensive details on this.

-L190: I would call the bias correction method a post-processing method rather than a downscaling method, to avoid confusions with downscaling methods used to refine the information granularity.

→ We agree with you and will replace downscale with post-processing.

-L210: Please provide the range of CRPS values. Additionally, at zero, the performance of the SFF would be considered the same as that of the ESP, so there would be no skill associated. Please clarify.

→ We will include additional information in the manuscript about the range of CRPS values (from 0 to ∞) and the meaning of a zero CRPSS.

-L233-234: Are these objectives the ones used to generate the pareto front? Please clarify.

→ Yes, they are used to generate the Pareto front. We will clarify this in the manuscript.

-L233-239: How are the ensembles considered in equations 5 and 6? Is the ensemble median used?

→ We used the mean value of SSD and SVD across ensemble members. See also our reply below to the comment about L.251-252.

-L244: Would it make more sense to calculate and present the forecast accuracy and skill for weekly aggregations rather than monthly, to match the aggregation periods of the SSD and SVD calculations?

→ We appreciate your suggestion; however, we disagree with presenting forecast accuracy and skill on a weekly basis. A monthly comparison, as shown in Figure 4, provides a more intuitive illustration of how mean error varies with different lead times.

-L251-252: It's unclear to me how various pareto fronts can be averaged. Are the individual solutions comparable across pareto fronts or is this an assumption? Please clarify.

→ Thank you for this comment. We realised that our original explanation in the manuscript was confusing. When optimising against ensemble forecasts (i.e. ESP or SFFs), the two objective functions (Eqs. 5 and 6, i.e. SSD and SVD) are evaluated against each ensemble members, and the average is taken as the final objective value and passed on to the NSGA-II optimiser. We will clarify this sentence in the manuscript.

-L254: I thought there were one million solutions, as per L249-250?

→ No, the number of solutions on the Pareto front is 100. We realised though that L249-250 are confusing and will reformulate them as: "We set the number of solutions to be evolved by the NGSA-II algorithm (so called "population" size) to 100, and the number of iterations to 100000, leading to a total of ten million model evaluations for each optimisation run"

-Figure 3: I would suggest writing out the acronyms (e.g., MCDM, WCD, etc.) in a table footer or in the caption so that the table could be understood as a standalone item.

→ Thank you for this suggestion and we will modify the figure.

-L337-338: Could you give us an indication of, for example, the spread of values and the mean per lead time? Here, interestingly the overall skill increases with increasing lead time. Could you infer some reasons for the skill increasing or decreasing with lead time for the various events and reservoirs in the results?

→ Yes, we will include the mean overall skill for each lead time in the manuscript, which decreases from 54% to 53% and 46% for 2, 4, and 6 months of lead time, respectively. This specific case of Soyanggang-Chungju for 2014-2016 was unique, and we were unable to identify clear reasons for its exceptional performance.

-Figure 5:

->It might be more coherent with section 3.1.2 to show the storage volume deficit instead of the storage volume.

→ The primary reason for displaying the storage volume instead of the storage volume difference (SVD) is its greater physical interpretability within the context of reservoir operation. We also believe that the SVD can be readily found from the current figure. To enhance clarity, we will incorporate a visual representation of the SVD into the figure (see the modified Figure 5 on page 7).

-> Could you label the dotted red-ish line at the top of each storage volume plot?

→ It represents the storage capacity (Smax.), and we will add a label in the figure (see the modified Figure 5 on page 7).

-L355: Was the wet event captured by the SFF? Knowing this could help explain some of the behaviours we can see in Fig. 5.

→ Yes, the SFFs captured the wet event in July more accurately than deterministic scenarios, as illustrated in Figure R2(a) below, and we agree that this help explains the better reservoir operations performance seen in Fig. 5. We will clarify this point in the manuscript and include this figure in the supplementary material for further reference

[Figure]

**Figure R2. (a) Cumulative flow observation (black square) and forecasts for Soyanggang-Chungju from June to July 2016, using WCD (orange diamond), 20YD (blue diamond), and SFFs (ensemble: hollow green circle, median: red circle). The black square represents the observed cumulative inflow during the same period. (b) Weekly demand and release, following the same colour coding and time period as in (a).**

-L355-357: Please expand on how we can see that the deterministic scenarios offer slightly superior results for securing storage volume compared to the ensemble forecasts on the figure. E.g., is the reservoir replenished faster? However, if the SFF knew that there was a rainfall event coming up, couldn't we expect that it recommends filling up the reservoir later to avoid losses linked with an overestimation of the storage by the end of the water year? Then, it wouldn't be fair to say that the deterministic scenarios offer superior results to secure storage volume over the SFF if the reservoir is fuller faster. Please expand on this in your results.

→ We agree your comment. Since SFFs predict wet future event more accurately than deterministic scenarios, they tend to release more water as seen in Figure R2 (b). We will clarify this point in the manuscript and ensure that we do not imply that deterministic scenarios yield superior results in securing storage volume over the SFFs. Additionally, we will also present the mean storage volume at the end of simulation period across all 48 simulations as shown in Figure 5.

[Figure]

Figure 5(Modified): Simulated reservoir operation results for Soyanggang-Chungju from June 2014 to September 2016 in terms of (a) storage volume and (b) cumulative squared supply deficit. From top to bottom, the rows represent simulation results generated by using WCD (orange), 20YD (blue), ESP (pink) and SFFs (green), respectively. Each sub-figure has 48 simulated results (coloured lines, 3 lead times × 8 MCDM methods × 2 decision-making time steps) and a single historical operation (black line). The numbers indicated at the right end of Figure 5(a) represent the mean storage volume (million m³) across all 48 simulations at Z (October 1st 2016).

-L371: Could the circles count be included somewhere in the text, figure or in a table?

→ Yes, we will add the circles count in the Figure 6 as shown below.

[Figure]

**Figure 6(Modified): Difference in SSD (x-axis) and SVD (y-axis) between historical operation (black cross) and simulated operations using different flow forecasts/scenarios (coloured circles) in Soyanggang-Chungju during the 2014-2016 drought. Performances are calculated on September 30th in (a) 2014, (b) 2015 and (c) 2016. Each sub-figure shows 48 points for each flow forecast/scenario (WCD, 20YD, ESP, SFFs), resulting from different combinations of key experimental choices (3 lead times × 8 MCDM methods × 2 decision-making time steps).**

-L380-388: "as the impact of forecast-informed operations accumulates" hints that the value of model-based "dynamic" forecasts has the potential to be even greater for longer drought events. This is a really interesting finding that I think would be nice to include in the discussion.

→ Thank you for this comment. We will add this in the manuscript.

-L389: Could the sensitivity results also be impacted by the different sample sizes of the experimental choices? Bootstrapping could help characterize some of the results' uncertainty.

→ We thank you for this comment. Following your advice, we applied bootstrapping technique for each experimental choice. We tested the sample size 20 with 3000 iterations. As shown in Figure R3 the results show that the impact of sample sizes to sensitivity is relatively small. We will include this figure in the supplementary material.

[Figure]

**Figure R3: (first row) Forecast value (y-axis) against key experimental choices: (a) forecast lead time, (b) MCDM method, (c) decision-making time step and (d) type of flow forecast/scenario for Soyanggang-Chungju reservoir system on September 30th, 2016. (Same figure as Figure 7 in the manuscript). (second row) Bootstrapped forecast value with 20 sample sizes and 3000 iterations.**

-L396-397: I think that the forecast value here refers to gains both in terms of the SSD and the SVD, but please remind readers here. Please also remind us here what the benchmark is.

→ Yes, you are right. We will add more explanations in this sentence.

-Figure 8: Should the dates in the legend be September 30th instead of October 1st, to match the legend of Fig. 7?

→ We agree with your point and will modify the legend of Figure 8 and Figure 9.

-L423: Can you make any educated guess with regards to why there is a lot of variability in the MCDM method results with events and reservoir systems?

→ We thank you for this comment. We hypothesize that the value can be influenced by the MCDM method, as well as the characteristic of analysed drought events.

When the Pareto front is plotted with the options selected by different MCDM methods, a distinct decision-making trend can be found (Figure R4). Except for the variable weighting method (Simple Selective, Multi Weighting), which applies different weights based on storage volume status, other methods demonstrate consistent decision-making trend and order as illustrated in Figure R4. The method emphasizes storage availability most significantly with the Storage-prioritized approach, followed by the Utopian point, TOPSIS, Balanced, Knee point, and, finally, the Supply-prioritized approach, as illustrated from right to left on the x-axis.

[Figure]

**Figure R4. Examples of the Pareto front and decision-making results based on different MCDM methods for Soyanggang-Chungju (a) and Andong-Imha (b) in June 2014.**

Taking into account this decision-making characteristic of each MCDM method, we reordered the MCDM methods (x-axis) from Storage-prioritized (Sto), Utopian point(UP), TOPSIS(TOP), Balanced(Bal), Knee point(KP) and Supply-prioritized (Sup), and isolated two Variable Weighting methods (SS, MW) as shown in Figure R5.

[Figure]

**Figure R5: (First row) Forecast value (y-axis) at the end of different drought events (points at Z, K, Z, M in Figure 1) plotted against MCDM methods. The methods are ordered from left to right with increasing importance to supply deficit (hollow circles), along with two variable weighting methods (SS and MW, black squares). (Second row) Same as first row in the middle of drought event (points at Y in Figure 1). Here, the lines are not intended to imply continuity; they are included solely to clarify the direction for visualization purposes.**

Figure R5 shows that, in the Soyanggang-Chungju reservoir system, the forecast value increases as the MCDM method prioritizes supply. In contrast, the opposite trend is observed in Andong-Imha. This discrepancy is closely linked to the characteristics of the drought events, as illustrated in Figure R6 below.

[Figure]

**Figure R6: Daily reservoir operation records for the studied drought events (K-water, 2023). Points X, Y, Z, J, K, L, M represent the ends of the hydrological years (September 30th) which will be used as points for forecast value assessment.**

For instance, the more emphasis placed on supply deficit, the higher the value we can achieve when the drought event concludes with a significant wet event (Soyanggang-Chungju, Figure R6(a, b)). This is because the effect of weighting storage availability is counteracted by the natural replenishment of storage from a wet event that occurs at the end of the simulation period. In contrast, MCDM methods that emphasize storage availability tend to achieve higher values when the drought event continues (Andong-Imha, Figure R6(c, d)).

We will address this in Section 5 'Discussion' and add the figures (R4-6) in the manuscript and supplementary material.

-L427-429: Why are we seeing those differences in the forecast value between the two regions? Does that somehow correlate with the skill of the seasonal meteorological forecasts in those regions or with how decisions were made historically? And what could explain the higher value of the SFF for the earlier event in the Soyanggang-Chungju reservoir system?

→ This is an important point. However, since we analysed only two reservoir systems and three drought events, it is also difficult to clearly explain why there are performance gaps between the two reservoir systems. We believe that further studies are required to clarify this issue. This limitation is discussed in section 5.2 Limitations and directions for future research (L526-530).

-L429-430: Why does increasing lead time lead to higher value?

→ We infer that this is due to the longer horizon of reliable future flow, which can provide operational benefits. However, the relationship between lead time and value is not strong, as indicated by the flow scenarios/forecasts (Figure 8(d)), and one of our results (Soyanggang-Chungju, 2016) shows an opposite trend. Therefore, we first need to clarify their relationship through further studies and then investigate the underlying reasons. We will clarify this point in the manuscript in Discussion section.

-L434-435: Please clarify in the text (and in the caption) that the y axis shows the value tallied over the 8 MCDM methods.

→ Yes. We will clarify this in the manuscript and caption.

-Figure 9: I think that the lines are a bit distracting in this figure. Could they be removed, with four different symbols used instead to represent the different events and reservoir systems?

→ We will modify the figure as shown below

[Figure]

**Figure 9(Modified): Relationship between forecast accuracy (Mean Error of monthly flow, x-axis) and value tallied over the 8 MCDM methods (y-axis) at the end of the simulation period for different drought events (2002, 2009 and 2016) at Soyanggang-Chungju and Andong-Imha reservoir systems. For each event and system, the figure shows five points corresponding to simulated forecast-informed operations using different forecasts/scenarios (orange: WCD, blue: 20YD, pink: ESP, green: SFFs, red: perfect forecast). The perfect forecast scenario was generated using actual flow observations as future forecasts. The direction of the blue arrows indicates higher performance (high value, low error), and the grey dashed lines represent the general expectation on the relationship between forecast accuracy and value.**

-L444: Please explain what the "perfect forecast" is in the caption and/or early on in the text.

→ We will include explanations of the perfect forecast both at the beginning of the text and in the caption (see the modified Figure 6 as shown above).

-L475: Except for the Soyanggang-Chungju earlier event.

→ The results from Soyanggang-Chungju reservoir system also show no significant difference in value between ensemble forecasts (ESP and SFFs). To avoid confusion, we will improve this sentence.

-L497: I don't understand why the method that prioritizes storage (over supply?) is more suitable for high risks linked with supply deficit. Could you please elaborate a bit for readers not as familiar with reservoir management?

→ The storage-prioritized method typically results in smaller supply deficits over longer periods, which helps prevent extreme storage shortage. Conversely, the supply-prioritized method tends to supply more than the storage-prioritized method. As a result, it carries a higher risk of extreme supply deficits over shorter periods when storage levels fall significantly, potentially leading to operational failure. We will add more explanations in the manuscript.

-L503-508: This is a repetition of the first discussion paragraph. Please consider combining both.

→ We agree with your suggestion. We will relocate this paragraph to follow the first paragraph in the discussion section and revise it to avoid repetition.

Technical corrections:

-L181: "analyzing" instead of "comparing"? → We will modify this.

-L249: Pareto. → We will modify this.

-L335: "outperforms" instead of "outperforming". → We will modify this.

-L406: "forecast" instead of "forecasts". → It will be modified.

**References**

Johnson, F. and Sharma, A. (2012). A nesting model for bias correction of variability at multiple time scales in general circulation model precipitation simulations. *Water Resources Research*, 48(1). doi:https://doi.org/10.1029/2011wr010464.

Lee, Y., Pianosi, F., Peñuela, A. and Rico-Ramirez, M. A.: Skill of seasonal flow forecasts at catchment-scale: an assessment across South Korea, *Hydrology and Earth System Science*, 28, 3261–3279, https://doi.org/10.5194/hess-28-3261-2024, 2024.

Maraun, D., Wetterhall, F., Ireson, A.M., Chandler, R.E., Kendon, E.J., Widmann, M., Brienen, S., Rust, H.W., Sauter, T., Themeßl, M., Venema, V.K.C., Chun, K.P., Goodess, C.M., Jones, R.G., Onof, C., Vrac, M. and Thiele-Eich, I. (2010). Precipitation downscaling under climate change: Recent developments to bridge the gap between dynamical models and the end user. *Reviews of Geophysics*, 48(3). doi:https://doi.org/10.1029/2009rg000314.

---

## Author Comment (AC2)

**Reply To Referee#2 Comments**

We thank referee for the valuable comments. Our replies to all comments are shown in blue and the original referee's comments are shown in black.

**Summary**: this manuscript attempts to show the value of seasonal flow forecasts for reservoir management by applying a simulation/optimization scheme and multicriteria decision making (MCDM) techniques to a reservoir system in South Korea. Seasonal Streamflow Forecasts (SSF) are used to run the model, and the performance is compared to that achieved by ESP and two deterministic scenarios consisting on the worst-case and 20yr observed droughts. The authors propose a metric to evaluate forecast value, consisting on the count value of times a forecast-simulation scenario outperforms the historical operation of the system. Then they evaluate the sensitivity of this forecast value to a series of methodological choices, including forecast lead time, MCDM technique, frequency of decision making, and type of streamflow forecast (deterministic or probabilistic) used.

**General comments**:

1. The authors tackle an important topic such as optimizing reservoir operations considering ever-increasing pressure on water resources and a less predictable hydrological regime due to climate change. The manuscript is well crafted, with attractive figures and an agile style which makes it enjoyable to read.

→ We sincerely appreciate your efforts in providing valuable feedback.

2. However, I am unconvinced that the material presented supports many of the claims and generalizations the manuscript makes. A fundamental concern is that of experimental design. The paper studies two reservoirs and three drought events, in what can hardly be considered a sample large enough to support claims about robustness.

→ We totally agree that our findings are limited to few reservoirs and drought events and thus we cannot draw robust general conclusions based on these limited cases. However, this is a limitation somehow 'intrinsic' to this type of simulation studies due to the rare occurrence of extreme drought events and the limited temporal cover of seasonal forecast data, which only became available since 1993. Since our results have shown the dependency of forecast value on reservoir systems and events, it is crucial to continue further testing to establish more general patterns in the skill-value relationship (if they exist), as well as to compare the performance between different forecast products. We hope that sharing our workflow and open-source code will stimulate and facilitate such further research.

In response to your comment, we will include these considerations as the very first point of sub-section 5.2 (Limitations and directions for future research) in the Discussion. We will also modify the title of the paper to 'Exploring the value of seasonal flow forecasts for drought management in South Korea' and clarify throughout the manuscript that such exploration is still based on a limited number of events and does not give a comprehensive answer yet regarding the skill-value relationship.

3. It appears that the events included are in all cases hydrological droughts with return periods of less than 20 years, because the 20-year drought always underestimates monthly flows. If this is actually the case, the experimental setup is predetermined to favor the ensemble over the deterministic methods, as the latter will underestimate flows and generate a poorer reservoir performance (evaluated ex-post).

→ Thanks for this comment, which made us realise that Figure 4 may be causing a misunderstanding regarding the magnitude of the 2014-2016 drought event. As shown in Figure R1 below, the cumulative reservoir inflows fell between the 20-year return period and the worst-case drought inflows for nearly two years, from June 2014 to May 2016 (it is only one out-of-season wet event in June 2016 that made the cumulative inflow catch up with and temporarily exceed the 20-year return period scenario). Given that the reservoirs in South Korea are designed to supply water for a year under a drought with a 20-year return period, this event is significant for practical reservoir operations. Since our simulation began in June 2014 and ran continuously until the end of 2016, it captures the impact of this severe drought event. To enhance the clarity regarding the magnitude of this drought event, we will include Figure R1 in the supplementary material of the revised manuscript.

[Figure]

**Figure R1: Cumulative flow observation (black dashed line) and forecasts (coloured solid lines) for Soyanggang-Chungju reservoir system for 2-month lead time. This figure corresponds to Figure 4(a) with 2-months lead time.**

4. Concerning the historical operation of the system, from the example shown in figure 5 it appears that reservoir operators did not realize they were facing a drought event until Jun-Jul 2015. After that date they accumulate large supply deficits, which can be more moderate when some forecast information allows hedging at the beginning of the simulation period (late 2014).

→ We fully agree with your point. As you mentioned, reservoir operators did not recognise the impending drought until Jun-July 2015, which exacerbated the damage caused by the drought. This operational failure highlights the limitations of the current reservoir operations in South Korea and further motivate our study to enhance our reservoir operation methodology utilising seasonal forecasts.

5. The discussion section mostly indicates that the results presented here confirm or align well previous research by some of the coauthors or by other groups. It is not easy to determine what is the main, novel, contribution of this paper to the wider body of literature, aside from the methodology for evaluating forecast value based on the count of outperforming scenarios.

→ We agree that some of our key findings well align with (few) previous studies that explore the topic with a similar approach to ours (i.e. quantifying the value of seasonal forecasts by simulating their use in historical events). We believe we have provided some useful contributions though: a methodology to measure value (as recognised by the author); a workflow to assess the sensitivity of simulation results to key set-up choice (something that was never done before, to our knowledge, and is both a methodological contribution for others to use, and provide interesting insights on which of those choices are very important and which are not); an attempt at beginning to systematise simulation experiments by analysing multiple reservoirs in the same region and multiple drought events - an incomplete attempt, admittedly, but a start to overcome the "single case study" approach of previous papers.

6. I would suggest that the authors take advantage of the fact that they have a limited sample of cases and analysed them in more detail: what are the hydrological characteristics of the drought events under study? How is the interplay between the temporality of monthly flows and demand? Are these reservoirs operated under multi-year criteria? Can operating rules (hedging) be introduced in combination with streamflow forecasts? In this way, they could glean further insights from their experiments. In the current version, many of the claims made in the discussion seem speculative.

→ We appreciate your suggestions. We think that these suggestions such as hydrological characteristics of drought events could be useful to improve our paper and will include those results in the manuscript. For example, Figure R1 will be included in the revised manuscript, accompanied by an explanation of the reservoir operation designed to ensure a single year water supply during 20-year return period drought events.

**Specific comments:**

L91. How is the material presented here relevant to the wider community outside of South Korea? All insights that are scalable to other systems should be highlighted.

→ Thank you for this comment. We will highlight the insights of our study to the wider research community outside of South Korea in Section 6 (Conclusions) by underlining that 'This study explores the potential of SFFs in improving drought management, and our workflow, accompanied by open-source code, will facilitate the replication of this analysis in other regions around the world'.

L112. This contribution does not seem substantial enough to justify the publication of this paper in its current form.

→ We thank you for this comment. While proposing a new methodology to quantify forecast value (L112) is very important point of our study, the key contribution of this study lies in exploring the forecast value of seasonal forecasts for enhancing drought management and providing our workflow for further research across other regions. We will further clarify this in the manuscript.

L132. It is said that both reservoirs operate as one, but later in the results section there are shown separately. Why?

→ This is incorrect. All the results shown in this manuscript pertain to the Soyanggang-Chungju and Andong-Imha reservoir systems as combined entities rather than as individual reservoirs. Although Soyanggang and Chungju are individual reservoirs, they are managed through conjunctive (combined) operation. Taking into account this, we have analysed them as Soyanggang-Chungju reservoir system. (This also applies to Andong and Imha reservoirs.)

L150: Here it is said that PET was computed, but the Penman-Monteith (FAO) metod computes real ET. This was confusing. Also, it is mentioned that evaporation from the reservoirs was neglected. This was surprising, because once you are running a simulation model, adding an empirical evaporation estimate shouldn't be too challenging. What was the reason to neglect this evaporation term?

→ We disagree with this comment. The Penman-Monteith method has been widely used to compute PET (e.g. Cai et al., 2007; Córdova et al., 2015). Additionally, we have neglected the evaporation from a reservoir due to the lack of relevant empirical equations in South Korea regarding the reservoir evaporation. This is discussed in 5.2 'Limitations and directions for future research' (L.531-538).

L304: replace "simply" with "simple".

→ We will replace this.

L315: It is difficult in the present manuscript to identify such insights.

→ This comment is also related to the robustness of the study. Please see the reply to the second comment in 'General comments' on page 1. We believe that this issue can also be addressed more by clearly articulating our objectives, contributions, and limitations.

L361: I don't think it is possible to say that "generally" something is true in this context, because the number of scenarios analyzed is very limited.

→ This is also related to the robustness issue. We will remove 'generally' here and clarify our objectives and contribution as the new attempt for assessing the value of seasonal forecasts for drought management.

L406. One problem of using the scenario count as forecast value is that it does not take into account the magnitude of the deficits generated in either of the scenarios tested.

→ Thank you for this comment and we agree with your comment. We also acknowledge that our methodology cannot measure the quality of the value. We think that incorporating the hypervolume concept, which defined as the space enclosed by a set of points in a multi-dimensional space (While et al., 2006; Sanchez-Gomez et al., 2019) could enhance our method for measuring the magnitude (or quality) of the value. Therefore, we will further discuss this in Section 5.2 'Limitations and directions for future research'.

L422-431. Here we see various results, but little analysis goes into gleaning the reasons why the MCDM method impacts differently the forecast value for either reservoir and drought. Before it was said that both reservoirs are operated jointly, but the results presented here suggest otherwise. Please clarify.

→ We thank you for this comment. We hypothesize that the value can be influenced by the MCDM method, as well as the characteristic of analysed drought events.

When the Pareto front is plotted with the options selected by the different MCDM methods, a distinct decision-making trend can be found (Figure R2). Except for the variable weighting method (Simple Selective, Multi Weighting), which applies different weights based on storage volume status, other methods demonstrate consistent decision-making trend and order as illustrated in Figure R2. The method emphasizes storage availability most significantly with the Storage-prioritized approach, followed by the Utopian point, TOPSIS, Balanced, Knee point, and, finally, the Supply-prioritized approach, as illustrated from right to left on the x-axis.

[Figure]

**Figure R2. Examples of the Pareto front and decision-making results based on different MCDM methods for Soyanggang-Chungju (a) and Andong-Imha (b) in June 2014.**

Taking into account this decision-making characteristic of the MCDM method, we reordered the MCDM methods (x-axis) from Storage-prioritized (Sto), Utopian point(UP), TOPSIS(TOP), Balanced(Bal), Knee point(KP) and Supply-prioritized (Sup), and isolated two Variable Weighting methods (SS, MW) as shown in Figure R3.

[Figure]

**Figure R3: (First row) Forecast value (y-axis) at the end of different drought events (points at Z, K, Z, M in Figure 1) plotted against MCDM methods. The methods are ordered from left to right with increasing importance to supply deficit (hollow circles), along with two variable weighting methods (SS and MW, black squares). (Second row) Same as first row in the middle of drought event (points at Y in Figure 1). Here, the lines are not intended to imply continuity; they are included solely to clarify the direction for visualization purposes.**

Figure R3 shows that, in the Soyanggang-Chungju reservoir system, the forecast value increases as the MCDM method prioritizes supply. In contrast, the opposite trend is observed in Andong-Imha. This discrepancy is closely linked to the characteristics of the drought events, as illustrated in Figure R4 below.

[Figure]

[Figure]

**Figure R4: Daily reservoir operation records for the studied drought events (K-water, 2023). Points X, Y, Z, J, K, L, M represent the ends of the hydrological years (September 30th) which will be used as points for forecast value assessment.**

For instance, the more emphasis is placed on supply deficit, the higher the value is achieved when the drought event concludes with a significant wet event (Soyanggang-Chungju, Figure R4(a, b)). This is because the effect of weighting storage availability is counteracted by the natural replenishment of storage from a wet event that occurs at the end of the simulation period. In contrast, MCDM methods that emphasize storage availability tend to achieve higher values when the drought event continues (Andong-Imha, Figure R4(c, d)).

We will address this in Section 5 'Discussion' and add the figures (R4-6) in the manuscript and supplementary material.

※   It is not true that our results show reservoirs separately. Please see reply to L132.

Figure 9: please clarify how is the "general expectation" curve obtained.

→ This curve conceptually illustrates the common understanding that higher forecast accuracy may lead to improved operational value. We will clarify this in the manuscript.

L461-462: please provide examples of the release scheduling policies derived by the different methods, in order to substantiate the idea of cautious operation. Also, please clarify the idea behind the concept "adverse events not seen during the optimization".

→ We can find the example of cautious operation before the wet event occurred in July 2016. Figure R5 shows that the flow forecast (a) and release decision (b) based on SFFs and deterministic scenarios (Worst Case Drought and 20 Year return period Drought scenarios). Considering flow uncertainty for the next two months, SFFs pertains higher releases compared to deterministic scenarios which predict continued dry conditions. Here, 'adverse events not seen during the optimization' can be the wet event

occurred in July. We will add more details in the manuscript and include Figure R5 in the supplementary material.

[Figure]

**Figure R5. (a) Cumulative flow observation (black square) and forecasts for Soyanggang-Chungju from June to July 2016, using WCD (orange diamond), 20YD (blue diamond), and SFFs (ensemble: hollow green circle, median: red circle). The black square represents the observed cumulative inflow during the same period. (b) Weekly demand and release, following the same colour coding and time period as in (a).**

L490. The logic behind this sentence is not evident.

→ Here, we aimed to highlight the importance of considering the uncertainty in flow forecasts. Specifically, while Figure 8(b) shows no clear relationship between the MCDM method and forecast value, Figure 8(d) indicates that the use of ensemble forecasts (i.e. considering the uncertainty in flow forecast) consistently provides higher value. We will revise this sentence to ensure a stronger connection to Figure 8, enhancing readers understanding.

L496-502. This is mostly speculation, which I suggest avoiding.

→ We believe that these sentences are grounded in practical considerations rather than speculation. For example, using the supply-prioritized method must be a good choice for a reservoir operation with ample storage capacity but lower demand. These sentences are significant as they offer recommendations for potential users in selecting an appropriate MCDM method, taking into account the primary purpose and operational characteristics of the reservoir.

L504-505. Why is this?

→ Even with perfect forecast of future inflows, limited lead times restrict our ability to consider subsequent conditions. As illustrated in Figure R6, if we use a lead time of 2-month and already recognize very dry conditions during this period, decisions will be made based solely on that dry condition. However, it is crucial to note that even a perfect forecast cannot account for wet events occurring beyond the 2-month horizon. We will revise the manuscript to explain this in more detail.

(a) Flow forecasts and release decision at t                   (b) Flow forecasts and release decision at t+Δt

[Figure]

[Figure]

**Figure R6. Conceptual illustration of the flow forecasts and release decision-making using a perfect forecast (upper row) and SFFs (lower row) at time t (a) and t + Δt (b).**

L527. I dispute the idea that the results presented here have demonstrated anything, because the number of experiments is limited, and because no actual explanation has been provided for the mechanics of the obtained results.

→ We think this comment can be addressed by clarifying the main objectives of the paper, key contributions and the limitations of this study. Here, we will modify this paragraph saying that:

"Firstly, while we assessed forecast value across two reservoir systems and historical drought events, these case studies may not be sufficient to draw comprehensive conclusions. The primary reasons for the lack of study cases are the infrequent occurrence of extreme drought events and the limited availability of seasonal forecast data, which only became available in 1993. Since our results have shown the dependency of forecast value on reservoir systems and events, it is crucial to continue further application efforts to establish more general patterns in the relationship between accuracy and value, as well as to compare the performance between different forecast products. While our study faces challenges related to robustness, we expect that our open-source code developed through this study will enable potential users to easily replicate our experiments and validate our provisional results across other regions around the world."

**Reference**:

Cai, J., Liu, Y., Lei, T. and Pereira, L.S.: Estimating reference evapotranspiration with the FAO Penman–Monteith equation using daily weather forecast messages, Agricultural and Forest Meteorology, 145(1-2), 22–35, https://doi.org/10.1016/j.agrformet.2007.04.012, 2007.

Córdova, M., Carrillo-Rojas, G., Crespo, P., Wilcox, B. and Célleri, R.: Evaluation of the Penman-Monteith (FAO 56 PM) Method for Calculating Reference Evapotranspiration Using Limited Data, Mountain Research and Development, 35(3), 230. https://doi.org/10.1659/mrd-journal-d-14-0024.1, 2015.

Sanchez-Gomez, J., Vega-Rodríguez, M.A. and Pérez, C.: Comparison of automatic methods for reducing the Pareto front to a single solution applied to multi-document text summarization, Knowledge-Based Systems journal, 174, 123–136, https://doi.org/10.1016/j.knosys.2019.03.002, 2019.

While, L., Hingston, P., Barone, L. and Huband, S.: A faster algorithm for calculating hypervolume, IEEE Transactions on Evolutionary Computation, 10(1), 29–38, https://doi.org/10.1109/tevc.2005.851275, 2006.

---

## Author Response (AR1)

**Author response**

We thank the editor and referees for their careful reading and helpful comments. Our reply is given below. The line numbers correspond to the modifications done on the revised manuscript.

**Referee 1**

-A clarification of the methods is needed, especially regarding the lead times and the decision-making time steps. For example, how can a decision be made every two months (i.e., bimonthly) for a forecast with two months lead time? Including a graphic illustrating the timeline between the forecast generation and the last decision made for a single water year would be really helpful, I think.

→ To clarify the methods used in this study, we have revised the text and added the Figure S2 in the supplementary material.

-More reflection is needed on the plausible physical explanations for some of the results to add some depth. For example, see my comment on L427-429.

→ We agree with you and are aware that this is very important point. However, it was challenging to derive physical explanations from our results confined to 2 reservoir systems and three drought events. This limitation is discussed in 5.2 'limitations and directions for future research'. For further details, please refer to the answer for L427-429 on page 6.

-Please discuss the shortcomings associated with evaluating only two attributes of the forecast performance (i.e., accuracy and skill). Calculating more attributes, like correlation, variance and reliability, would give a fuller picture, which could impact the conclusion you draw on L448-450 regarding the link between forecast performance and value.

→ We have added a sentence in the Discussion to raise this point (L.555-560).

-Are the codes you developed for the evaluation shared anywhere for others to follow your approach more easily? If not, please consider making them available.

→ We have organized the code developed in this study for evaluation purposes. It is now publicly available as part of the iRONS python package (https://ironstoolbox.github.io/). We have added the link to access our code in the revised manuscript (L.652-653).

-L23: In the abstract, please specify what key choices you're looking at.

→ We have specified the types of key choices in the abstract (L.22-23).

-L113-114: I would move this last sentence to the conclusions instead as it seems a bit out of place in the introduction.

→ We have removed the last sentence from the Introduction and improved the Conclusions.

-L122-130: Could you give a brief description of the hydrological regime of both regions? E.g., When are the peak flows? What drives runoff generation?

→ We have included a brief description of the hydrological regime of reservoir systems (L.138-141).

-L140: On L39 the dates for this event are 2013-2016. Please clarify.

→ To avoid confusion, we have standardised the drought period as 2014-2016 in the manuscript, aligning it with our simulation period (L.40 and L.145).

-L159-162: What is the initialization frequency of the forecasts and what time period is covered by the forecasts you generated for this study?

→ We have added details of the initialization frequency in the manuscript (L.169-172).

-L162-163: The time period based on which the correction factors are calculated overlaps with the drought events in 2001-2002 and 2008-2009. This could be an unfair advantage for these events. Please clarify. Same comment for the ESP generation explained on L186.

→ We used the time period 1993-2010 for generating the bias correction factors because of concerns about data sufficiency. For example, when analysing drought events from 2001 to 2002, only 7 years of data from 1993 to 2000 would be available if we tried to avoid overlap with the event period. We agree that incorporating overlapped years could potentially provide unfair advantages, but using an insufficient amount of data for generating bias correction factors can also lead to significant issues. Johnson and Sharma (2012) and Maraun et al. (2010) suggest that larger datasets help ensure more accurate bias corrections by capturing the variability of the data better and reducing the influence of outliers. By fixing the time period from 1993 to 2010, we ensure a more reliable and robust calculation. We have applied the same time period constraint to the ESP calculation until 2010 to maintain consistency of our study. We have clarified this in the manuscript (L.213-217).

-L167: Can you briefly list the four different forecasts/scenarios here as well?

→ We have added brief list of the forecasts and scenarios (L.177-178).

-L172-173: What is the temporal aggregation of the forecasts/scenarios, based on which the decisions are made? E.g., Weekly, monthly, etc.

→ We have clarified the temporal aggregation details in the manuscript (L.196-200).

-L173: Please specify how much time there is in between each decision.

→ Please, see our response to L174 for Referee 1 shown below.

-L174: Is the process iteratively conducted at the start of each month? The frequency is unclear.

→ This reservoir optimisation and simulation process is iteratively conducted at the start of every month (for monthly decision-making) or every two months (for bimonthly decision-making) throughout the simulation period (e.g. Jun. 2014 to Sep. 2016). This is clarified this in Figure S2 in the supplementary material and revised manuscript (L.198-200).

-Figure 2: Very nice graphic! You could refer the readers to each figure compartment being described at the start of each sub-section below.

→ We have added this information in the manuscript (L.176).

-> Could you specify whether bimonthly refers to twice a month or every two months please?

→ We have modified the Figure 3, and this is clarified in Figure S2.

->It would be useful to add a short section (3.1.4) for the reservoir simulation step, both to have coherence between the numbering of the sections and the boxes in the figure (i.e., step 1 is explained in 3.1.1, step 2 in 3.1.2, etc.) and to provide some information on how this is done (e.g., I don't quite understand how decisions are made at different time steps with forecasts that cover different lead times and how often a new forecast is produced).

→ We have added section 3.1.4 for the reservoir simulation step (L.321-331).

-L178-193: Please provide more information about the forecasts' generation, with regards to the: simulation periods, forecast time steps, initialization dates, lead time (please also explain how you define lead time with a concrete example as different research groups define them differently, e.g., lead month 0 vs. 1), ensemble size for the SFF.

→ We have clarified the details in the manuscript (L.196-200, 211).

-L181-182: Operationally, with what lead times and at what time steps are the decisions made currently by K-water? This would help contextualize your methodological decisions.

→ We have included this information in the manuscript (L.141-143).

-L185-189: Could you comment on the difference in ensemble sizes between the ESP and the SFF and the potential impacts on the performance evaluation?

→ Followed by your comment (L178-193), the ensemble sizes of ESP (45) and SFFs (25 until 2016, 51 since 2017) have been included in the manuscript (L.207, L.211). However, evaluating the impact of the ensemble size is out of the scope of this study given the limited number of drought events. A previous study conducted by Peñuela et al. (2020) showed that when the number of ensembles decreases below 10, the forecast value can be affected.

-L186-187: Could you give a bit more information about the Tank hydrological model, such as its spatial resolution, how it was calibrated, how the initial conditions were obtained, and what its performance in simulation is for the basins considered here.

→ We have briefly added information about the model to this manuscript (L.207-209). Further details on the Tank model are documented in our previous paper as mentioned in the manuscript (L.217-219).

-L190: I would call the bias correction method a post-processing method rather than a downscaling method, to avoid confusions with downscaling methods used to refine the information granularity.

→ We have modified the sentence (L.212-213).

-L210: Please provide the range of CRPS values. Additionally, at zero, the performance of the SFF would be considered the same as that of the ESP, so there would be no skill associated. Please clarify.

→ We have included additional information in the manuscript about the CRPS (L.235-239).

-L233-234: Are these objectives the ones used to generate the pareto front? Please clarify.

→ Yes, they are used to generate the Pareto front. We have clarified this in the manuscript (L.279).

-L233-239: How are the ensembles considered in equations 5 and 6? Is the ensemble median used?

→ We used the mean value of SSD and SVD across ensemble members. See also our reply below to the comment about L282-283.

-L244: Would it make more sense to calculate and present the forecast accuracy and skill for weekly aggregations rather than monthly, to match the aggregation periods of the SSD and SVD calculations?

→ We think that a monthly comparison, as shown in Figure 4, provides a more intuitive illustration of how mean error varies with different lead times.

-L251-252: It's unclear to me how various pareto fronts can be averaged. Are the individual solutions comparable across pareto fronts or is this an assumption? Please clarify.

→ We have clarified this sentence in the manuscript (L.279-283).

-L254: I thought there were one million solutions, as per L249-250?

→ The number of solutions on a Pareto front is 100. We have reformulated this sentence (L.279-281).

-Figure 3: I would suggest writing out the acronyms (e.g., MCDM, WCD, etc.) in a table footer or in the caption so that the table could be understood as a standalone item.

→ We have improved Figure 4.

-L337-338: Could you give us an indication of, for example, the spread of values and the mean per lead time? Here, interestingly the overall skill increases with increasing lead time. Could you infer some reasons for the skill increasing or decreasing with lead time for the various events and reservoirs in the results?

→ In general, the overall skill decreases with increasing lead time. This specific case of Soyanggang-Chungju for 2014-2016 was unique, and we were unable to identify clear reasons for its exceptional performance. We added the explanation of the general trend shown in other reservoir systems and drought events in the revised manuscript (L.379-380).

-Figure 5: ->It might be more coherent with section 3.1.2 to show the storage volume deficit instead of the storage volume.

→ The primary reason for displaying the storage volume instead of the storage volume difference (SVD) is its greater physical interpretability within the context of reservoir operation. We also believe that the SVD can be readily found from the current figure. To enhance clarity, we have incorporated a visual representation of the SVD into Figure 6.

-> Could you label the dotted red-ish line at the top of each storage volume plot?

→ It represents the storage capacity (Smax.), and we have added a label in Figure 6.

-L355: Was the wet event captured by the SFF? Knowing this could help explain some of the behaviours we can see in Fig. 5.

→ We have clarified this point in the manuscript and included a figure in the supplementary material (Figure S7), demonstrating that the SFFs successfully captured the wet event in June 2016 (L.410-413).

-L355-357: Please expand on how we can see that the deterministic scenarios offer slightly superior results for securing storage volume compared to the ensemble forecasts on the figure. E.g., is the reservoir replenished faster? However, if the SFF knew that there was a rainfall event coming up, couldn't we expect that it recommends filling up the reservoir later to avoid losses linked with an overestimation of the storage by the end of the water year? Then, it wouldn't be fair to say that the deterministic scenarios offer superior results to secure storage volume over the SFF if the reservoir is fuller faster. Please expand on this in your results.

→ We have added the mean storage volume at the end of simulation period across all 48 simulations in Figure 6 and discussed this in the manuscript (L.404-413).

-L371: Could the circles count be included somewhere in the text, figure or in a table?

→ We have added the circles count in Figure 7.

-L380-388: "as the impact of forecast-informed operations accumulates" hints that the value of model-based "dynamic" forecasts has the potential to be even greater for longer drought events. This is a really interesting finding that I think would be nice to include in the discussion.

→ We agree and have included this in the manuscript (L.435-436).

-L389: Could the sensitivity results also be impacted by the different sample sizes of the experimental choices? Bootstrapping could help characterize some of the results' uncertainty.

→ We applied bootstrapping technique for each experimental choice. Specifically, we created 3000 bootstrap resamples, each with a size of 20. The results show that the impact of sample sizes to sensitivity is relatively small. We have included this result in the supplementary material as Figure S9 and discussed in the manuscript (L.470-472).

-L396-397: I think that the forecast value here refers to gains both in terms of the SSD and the SVD, but please remind readers here. Please also remind us here what the benchmark is.

→ We have included additional explanations in this sentence (L.446-447).

-Figure 8: Should the dates in the legend be September 30th instead of October 1st, to match the legend of Fig. 7?

→ We have modified the legend in Figure 9 and Figure 10.

-L423: Can you make any educated guess with regards to why there is a lot of variability in the MCDM method results with events and reservoir systems?

→ We have modified the Figure 8 (b) bottom row and Figure 9 (b) as an order of increasing importance to storage (see Figure S9). We then discussed the further relationship between the value and MCDM method in the manuscript (L.460-469, 486-494).

-L427-429: Why are we seeing those differences in the forecast value between the two regions? Does that somehow correlate with the skill of the seasonal meteorological forecasts in those regions or with how decisions were made historically? And what could explain the higher value of the SFF for the earlier event in the Soyanggang-Chungju reservoir system?

→ This is an important point. However, since we analysed only two reservoir systems and three drought events, it is also difficult to clearly explain why there are performance gaps between the two reservoir systems. We believe that further studies are required to clarify this issue. This limitation is discussed in section 5.2 Limitations and directions for future research (L.595-603).

-L429-430: Why does increasing lead time lead to higher value?

→ We infer that this is due to the longer horizon of reliable future flow, which can provide operational benefits. However, the relationship between lead time and value is not strong, and one of our results (Soyanggang-Chungju, 2016) shows an opposite trend (Figure 9(a)). Therefore, we first need to clarify their relationship through further studies and then investigate the underlying reasons. We have clarified this point in the manuscript (L.567-568).

-L434-435: Please clarify in the text (and in the caption) that the y axis shows the value tallied over the 8 MCDM methods.

→ We have clarified this in the manuscript and caption of Figure 10 (L.502, 510-511).

-Figure 9: I think that the lines are a bit distracting in this figure. Could they be removed, with four different symbols used instead to represent the different events and reservoir systems?

→ We have modified Figure 10.

-L444: Please explain what the "perfect forecast" is in the caption and/or early on in the text.

→ We have included explanations of the perfect forecast both at the beginning of the text and in the caption (L.506-508, 513-514).

-L475: Except for the Soyanggang-Chungju earlier event.

→ The results from Soyanggang-Chungju reservoir system also show no significant difference in value between ensemble forecasts (ESP and SFFs) (L.550-551).

-L497: I don't understand why the method that prioritizes storage (over supply?) is more suitable for high risks linked with supply deficit. Could you please elaborate a bit for readers not as familiar with reservoir management?

→ The storage-prioritized method typically results in smaller supply deficits over longer periods, which helps prevent extreme storage shortage. Conversely, the supply-prioritized method tends to supply more than the storage-prioritized method. As a result, it carries a higher risk of extreme supply deficits over shorter periods when storage levels fall significantly, potentially leading to operational failure. We have included additional explanations on this (L.577-579).

-L503-508: This is a repetition of the first discussion paragraph. Please consider combining both.

→ We have relocated this paragraph and modified it (L.543-549).

Technical corrections:

-L181: "analyzing" instead of "comparing"? → We have modified this (L.201).

-L249: Pareto. → We have modified this sentence (L.280-281).

-L335: "outperforms" instead of "outperforming". → We have modified this (L.377).

-L406: "forecast" instead of "forecasts". → We have modified this (L.463).

**Referee 2**

1. The authors tackle an important topic such as optimizing reservoir operations considering ever-increasing pressure on water resources and a less predictable hydrological regime due to climate change. The manuscript is well crafted, with attractive figures and an agile style which makes it enjoyable to read.

→ We sincerely appreciate your efforts in providing valuable feedback.

2. However, I am unconvinced that the material presented supports many of the claims and generalizations the manuscript makes. A fundamental concern is that of experimental design. The paper studies two reservoirs and three drought events, in what can hardly be considered a sample large enough to support claims about robustness.

→ We agree that our findings are limited to few reservoirs and drought events and thus we cannot draw robust general conclusions based on these limited cases. However, this is a limitation somehow 'intrinsic' to this type of simulation studies due to the rare occurrence of extreme drought events and the limited temporal cover of seasonal forecast data, which only became available since 1993. Since our results have shown the dependency of forecast value on reservoir systems and events, it is crucial to continue further testing to establish more general patterns in the skill-value relationship (if they exist), as well as to compare the performance between different forecast products. We hope that sharing our workflow and open-source code will stimulate and facilitate such further research.

In response to your comment, we have included these considerations as the very first point of sub-section 5.2 (Limitations and directions for future research) in the Discussion (L.595-603). We also modified the title of the paper to 'Exploring the value of seasonal flow forecasts for drought management in South Korea' (L.1-2) and clarified throughout the manuscript that such exploration is still based on a limited number of events and does not give a comprehensive answer yet to our "value" question.

3. It appears that the events included are in all cases hydrological droughts with return periods of less than 20 years, because the 20-year drought always underestimates monthly flows. If this is actually the case, the experimental setup is predetermined to favor the ensemble over the deterministic methods, as the latter will underestimate flows and generate a poorer reservoir performance (evaluated ex-post).

→ We realise that Figure 4 may be causing a misunderstanding regarding the magnitude of the 2014-2016 drought event. Therefore, we have compared the inflow observation and flow scenarios/forecasts and found that the magnitude of this event is between the 20-year return period and the worst-case drought for nearly two years, from June 2014 to May 2016. Given that the reservoirs in South Korea are designed to supply water for a year under a drought with a 20-year return period, this event is significant for practical reservoir operations. Since our simulation began in June 2014 and ran continuously until the end of 2016, it captures the impact of this severe drought event. To enhance the clarity regarding the magnitude of this drought event, we have added Figure S5 in the supplementary material and included additional explanations in the manuscript (L.380-384).

4. Concerning the historical operation of the system, from the example shown in figure 5 it appears that reservoir operators did not realize they were facing a drought event until Jun-Jul 2015. After that date they accumulate large supply deficits, which can be more moderate when some forecast information allows hedging at the beginning of the simulation period (late 2014).

→ We fully agree with your point. As you mentioned, reservoir operators did not recognise the impending drought until Jun-July 2015, which exacerbated the damage caused by the drought. This

operational failure highlights the limitations of the current reservoir operations in South Korea and further motivate our study to enhance our reservoir operation methodology utilising seasonal forecasts. Therefore, we have included more detailed explanation of what happened historically in the manuscript (L.398-401).

5. The discussion section mostly indicates that the results presented here confirm or align well previous research by some of the coauthors or by other groups. It is not easy to determine what is the main, novel, contribution of this paper to the wider body of literature, aside from the methodology for evaluating forecast value based on the count of outperforming scenarios.

→ We agree that some of our key findings well align with (few) previous studies that explore the topic with a similar approach to ours (i.e. quantifying the value of seasonal forecasts by simulating their use in historical events). We think reporting these 'confirmatory' results is still important given the very limited number of studies of this type so far. Moreover, we believe our manuscript provides some key novel contributions:

A. a workflow to assess the sensitivity of simulation results to key set-up choice (something that was never done before, to our knowledge, and is both a methodological contribution for others to use, and provide interesting insights on which of those choices are very important and which are not - at least in our study region);

B. a methodology to measure value in a way that acknowledges uncertainty in the simulation results due to experimental set-up choices while capturing trade-offs between the conflicting objectives in a simple, synthetic way;

C. an attempt at beginning to systematise the skill-value assessment by analysing multiple reservoirs in the same region and multiple drought events - an incomplete attempt, admittedly, but a start to overcome the "single case study" approach of previous papers.

We think these contributions are important as they can help the research community to move forward towards the search for "general principles" to understanding when and how SFFs bring value to reservoir operations optimisation. To this end, we need to ensure that simulation results are not (too much) dependent on set-up choices in our simulation experiments, and that we repeat experiments for different reservoirs/events within sufficiently homogeneous regions (to begin with).

This said, we agree with the Referee 2 that these contributions were not well articulated in our original manuscript, so we have introduced a clear statement of our contributions in 6. Conclusions (L.640-651).

6. I would suggest that the authors take advantage of the fact that they have a limited sample of cases and analysed them in more detail: what are the hydrological characteristics of the drought events under study? How is the interplay between the temporality of monthly flows and demand? Are these reservoirs operated under multi-year criteria? Can operating rules (hedging) be introduced in combination with streamflow forecasts? In this way, they could glean further insights from their experiments. In the current version, many of the claims made in the discussion seem speculative.

→ We have incorporated more details about the drought event and reservoir operations in the revised manuscript. Firstly, we have divided current Figure 1 into two figures (as Figure 1 and 2), so that we can show better the time series of the three drought events and give more details of their hydrological characteristics (L.115-122, 138-141, 380-384). Secondly, we have added more explanations and analysis results of the mechanisms that explain the differences in simulated operations shown in Figure 6 (L.410-413).

L91. How is the material presented here relevant to the wider community outside of South Korea? All insights that are scalable to other systems should be highlighted.

→ We have clarified the main contributions of this study, including the relevance to the wider community, in 6. Conclusions (L.641-645).

L112. This contribution does not seem substantial enough to justify the publication of this paper in its current form.

→ We have removed this sentence from the Introduction. As discussed in response to General Point 5 above, we have improved our major contributions in 6. Conclusions (L.640-651).

L132. It is said that both reservoirs operate as one, but later in the results section there are shown separately. Why?

→ We recognise that this sentence is confusing and have improved it to clarify the reservoir operations in the manuscript (L.132-138).

L150: Here it is said that PET was computed, but the Penman-Monteith (FAO) metod computes real ET. This was confusing. Also, it is mentioned that evaporation from the reservoirs was neglected. This was surprising, because once you are running a simulation model, adding an empirical evaporation estimate shouldn't be too challenging. What was the reason to neglect this evaporation term?

→ As we understand, the Penman-Monteith method has been widely used to compute PET (e.g. Cai et al., 2007; Córdova et al., 2015). Additionally, we have neglected the evaporation from a reservoir due to the lack of relevant empirical equations in South Korea. This is discussed in 5.2 'Limitations and directions for future research' (L.614-621).

L304: replace "simply" with "simple".

→ We have replaced this (L.346).

L315: It is difficult in the present manuscript to identify such insights.

→ As addressed in our response to General Point 2 (page 8), we have modified the title of the paper to 'Exploring the value of seasonal flow forecasts for drought management in South Korea', and also included these considerations as the very first point of sub-section 5.2 Limitations and directions for future research (L.1-2, 595-603).

L361: I don't think it is possible to say that "generally" something is true in this context, because the number of scenarios analyzed is very limited.

→ We have removed this sentence from the manuscript and included additional experiment results on the possibility of SFFs in predicting wet event (L.410-413).

L406. One problem of using the scenario count as forecast value is that it does not take into account the magnitude of the deficits generated in either of the scenarios tested.

→ We also acknowledge that our methodology cannot measure the quality of the value. We think that incorporating the 'hypervolume' concept, which defined as the space enclosed by a set of points in a multi-dimensional space (While et al., 2006; Sanchez-Gomez et al., 2019) could enhance our method for measuring the magnitude (or quality) of the value. We have discussed this in Section 5.2 'Limitations and directions for future research' (L.608-612).

L422-431. Here we see various results, but little analysis goes into gleaning the reasons why the MCDM method impacts differently the forecast value for either reservoir and drought. Before it was said that both reservoirs are operated jointly, but the results presented here suggest otherwise. Please clarify.

→ As Referee 1 also raised a similar point (see our respond to L423 for Referee 1), we have addressed this in detail in revised manuscript (L.460-469, 486-494).

Figure 9: please clarify how is the "general expectation" curve obtained.

→ This curve is conceptual illustration of the common understanding that higher forecast accuracy may lead to improved operational value. To clarify this, we have improved the caption of Figure 10.

L461-462: please provide examples of the release scheduling policies derived by the different methods, in order to substantiate the idea of cautious operation. Also, please clarify the idea behind the concept "adverse events not seen during the optimization".

→ We have added examples of release schedules and expand the text to better explains the reasons for the differences in behaviours across methods (L.410-413, Figure S7 in the supplementary material). For more details, please refer to the response to Referee 1 (L423) on page 5.

L490. The logic behind this sentence is not evident.

→ While Figure 9(b) shows no clear relationship between the MCDM method and forecast value, Figure 9(d) indicates that the use of ensemble forecasts (i.e. considering the uncertainty in flow forecast) consistently provides higher value. We have revised this sentence to ensure a stronger connection to Figure 9, enhancing readers understanding (L.567-571).

L496-502. This is mostly speculation, which I suggest avoiding.

→ We believe that these sentences are grounded in practical considerations rather than speculation. For example, using the supply-prioritized method must be a good choice for a reservoir operation with ample storage capacity but lower demand. These sentences are significant as they offer recommendations for potential users in selecting an appropriate MCDM method, taking into account the primary purpose and operational characteristics of the reservoir. To clarify this, we have improved this paragraph (L.575-583).

L504-505. Why is this?

→ To help the reader's understanding, we have added Figure S10 in the supplementary material to explain this more clearly (L.544-547).

L527. I dispute the idea that the results presented here have demonstrated anything, because the number of experiments is limited, and because no actual explanation has been provided for the mechanics of the obtained results.

→ We hope our responses above and revisions of the manuscript may convince the Referee of the usefulness of our results.

**Reference**:

Cai, J., Liu, Y., Lei, T. and Pereira, L.S.: Estimating reference evapotranspiration with the FAO Penman–Monteith equation using daily weather forecast messages, Agricultural and Forest Meteorology, 145(1-2), 22–35, https://doi.org/10.1016/j.agrformet.2007.04.012, 2007.

Córdova, M., Carrillo-Rojas, G., Crespo, P., Wilcox, B. and Célleri, R.: Evaluation of the Penman-Monteith (FAO 56 PM) Method for Calculating Reference Evapotranspiration Using Limited Data, Mountain Research and Development, 35(3), 230. https://doi.org/10.1659/mrd-journal-d-14-0024.1, 2015.

Johnson, F. and Sharma, A. (2012). A nesting model for bias correction of variability at multiple time scales in general circulation model precipitation simulations. *Water Resources Research*, 48(1). doi:https://doi.org/10.1029/2011wr010464.

Lee, Y., Pianosi, F., Peñuela, A. and Rico-Ramirez, M. A.: Skill of seasonal flow forecasts at catchment-scale: an assessment across South Korea, *Hydrology and Earth System Science*, 28, 3261–3279, https://doi.org/10.5194/hess-28-3261-2024, 2024.

Maraun, D., Wetterhall, F., Ireson, A.M., Chandler, R.E., Kendon, E.J., Widmann, M., Brienen, S., Rust, H.W., Sauter, T., Themeßl, M., Venema, V.K.C., Chun, K.P., Goodess, C.M., Jones, R.G., Onof, C., Vrac, M. and Thiele-Eich, I. (2010). Precipitation downscaling under climate change: Recent developments to bridge the gap between dynamical models and the end user. *Reviews of Geophysics*, 48(3). doi:https://doi.org/10.1029/2009rg000314.

Peñuela, A., Hutton, C. and Pianosi, F.: Assessing the value of seasonal hydrological forecasts for improving water resource management: insights from a pilot application in the UK, Hydrology and Earth System Sciences, 24(12), 6059–6073, https://doi.org/10.5194/hess-24-6059-2020, 2020.

Sanchez-Gomez, J., Vega-Rodríguez, M.A. and Pérez, C.: Comparison of automatic methods for reducing the Pareto front to a single solution applied to multi-document text summarization, Knowledge-Based Systems journal, 174, 123–136, https://doi.org/10.1016/j.knosys.2019.03.002, 2019.

While, L., Hingston, P., Barone, L. and Huband, S.: A faster algorithm for calculating hypervolume, IEEE Transactions on Evolutionary Computation, 10(1), 29–38, https://doi.org/10.1109/tevc.2005.851275, 2006